# Acoustic Biosensors and Microfluidic Devices in the Decennium: Principles and Applications

**DOI:** 10.3390/mi13010024

**Published:** 2021-12-26

**Authors:** Minu Prabhachandran Nair, Adrian J. T. Teo, King Ho Holden Li

**Affiliations:** School of Mechanical and Aerospace Engineering, Nanyang Technological University, Singapore 639798, Singapore; MINU001@e.ntu.edu.sg (M.P.N.); adrian.teojt@ntu.edu.sg (A.J.T.T.)

**Keywords:** BAW, SAW, QCM, FBAR, piezoelectric, biosensing, microfluidics, actuation, POC, LOC

## Abstract

Lab-on-a-chip (LOC) technology has gained primary attention in the past decade, where label-free biosensors and microfluidic actuation platforms are integrated to realize such LOC devices. Among the multitude of technologies that enables the successful integration of these two features, the piezoelectric acoustic wave method is best suited for handling biological samples due to biocompatibility, label-free and non-invasive properties. In this review paper, we present a study on the use of acoustic waves generated by piezoelectric materials in the area of label-free biosensors and microfluidic actuation towards the realization of LOC and POC devices. The categorization of acoustic wave technology into the bulk acoustic wave and surface acoustic wave has been considered with the inclusion of biological sample sensing and manipulation applications. This paper presents an approach with a comprehensive study on the fundamental operating principles of acoustic waves in biosensing and microfluidic actuation, acoustic wave modes suitable for sensing and actuation, piezoelectric materials used for acoustic wave generation, fabrication methods, and challenges in the use of acoustic wave modes in biosensing. Recent developments in the past decade, in various sensing potentialities of acoustic waves in a myriad of applications, including sensing of proteins, disease biomarkers, DNA, pathogenic microorganisms, acoustofluidic manipulation, and the sorting of biological samples such as cells, have been given primary focus. An insight into the future perspectives of real-time, label-free, and portable LOC devices utilizing acoustic waves is also presented. The developments in the field of thin-film piezoelectric materials, with the possibility of integrating sensing and actuation on a single platform utilizing the reversible property of smart piezoelectric materials, provide a step forward in the realization of monolithic integrated LOC and POC devices. Finally, the present paper highlights the key benefits and challenges in terms of commercialization, in the field of acoustic wave-based biosensors and actuation platforms.

## 1. Introduction

The inception of lab-on-a-chip (LOC) technologies gracefully amalgamated the use of bulky laboratory apparatus with the need for complex procedures. This brought forth a game-changing, miniaturized platform capable of highly automated operations [1,2]. Some examples include the transportation and subsequent mixing and splitting of biological samples for detection, and the conversion of biological recognition events into detectable electrical output signals [3,4,5,6]. LOC technology also plays a significant role in the development of point of care (POC) diagnostics, allowing complex laboratory procedures to be performed by end-users themselves with cheaper and easier access for all [7]. Some examples of these LOC technologies are label-free biosensors [8,9] and/or microfluidic actuation devices [10,11,12,13]. These have been the key enablers towards the development of such LOC and POC devices.

Biosensors, as the name implies, are devices that detect biomolecular recognition events through the use of a transducer, converting the physiochemical reaction phenomenon into a measurable electrical output [8,14]. They enable a real-time, label-free, simple and miniaturized platform for detecting a wide range of analytes from biomolecules, namely disease-causing biomarker antigens; proteins; pathogenic microorganisms; complex nucleic acid molecules; glucose monitoring; pH monitoring; and blood coagulation monitoring [15,16,17,18,19,20,21,22]. Two basic components are generally required for a biosensor, namely a bioreceptor layer that detects the biomolecules, and a transducer. Biosensors can then be categorized based on their transduction mechanisms into electrochemical, thermal, electrical, magnetic, optics, and piezoelectric [23].

Among the various biosensor transduction mechanisms, piezoelectric acoustic biosensors are the best as they enable the real-time, label-free, and rapid detection of analytes without the need for specific reagents and complex sample preparations [24]. They operate on the principle of detecting the label-free affinity reactions between the bioreceptor and target analytes utilizing the electromechanical conversion property of piezoelectric materials [25]. Acoustic biosensors use acoustic waves generated by piezoelectric materials to detect the target analyte via the induced variations in acoustic wave velocity, frequency, and amplitude [26]. Bulk piezoelectric ceramic substrates, as well as thin piezoelectric deposited films, are used for the generation of acoustic waves.

Microfluidics is a widely publicized LOC device that is used in multidisciplinary research. It is capable of providing advantages, such as lower costs with a high accuracy, due to its inherent nature of requiring low sample volumes with high throughput [27,28]. Various actuation mechanisms, such as capillary-driven, pressure-driven, electrokinetic, centrifugal, and acoustic methods, have been demonstrated to effectively manipulate fluid samples [29]. The acoustic method using piezoelectric materials is an especially promising candidate for biosensing and actuation applications. Unlike most methods that can directly affect the flow field across the whole device, selectivity of focus can be achieved using this active method of actuation, for all types of fluid medium [30,31,32,33,34,35]. Reliable sensing and actuation can also be achieved from the unique sensing frequencies for different biomolecules without destroying the biomolecules themselves [36,37]. One key advantage of piezoelectric-based acoustic technology found in LOCs and microfluidics is that both sensing and actuation can be realized using acoustic waves generated by piezoelectric materials [1]. This enables the development of an integrated platform. Microfluidics has been observed to be performed using closed channels and open fields (digital microfluidics, DMF). Both systems can be realized using piezoelectric materials with a simple design and ease of operation. Digital microfluidic devices, however, have an added benefit as they are a planar technology with no need for any external components for pumping or fluid dispensing compared to continuous flow [10].

Acoustic waves can be broadly categorized into two types, namely bulk acoustic waves (BAW) and surface acoustic waves (SAW), depending on the propagation of mechanical waves in the piezoelectric materials. Both BAW and SAW waves have different modes that can be utilized for biosensing as well as in microfluidic actuation. Various acoustic wave modes of BAW and SAW utilized for biosensing and microfluidic actuation are represented schematically in Figure 1a, which will be described in detail in each of the sensor categories. The schematic representation of the application of acoustic wave modes in sensing and actuation is shown in Figure 1b.

This review paper focuses on the use of BAW and SAW biosensors based on the piezoelectric materials for LOC and POC diagnostic tools. A brief introduction to the BAW- and SAW-based microfluidic actuation and its applications in LOC has also been included. The detailed study of the piezoelectric acoustic wave-based microfluidic actuation platforms is out of the scope of this review. Section 2 details the acoustic wave-based biosensors discussing, in further detail, BAW biosensors using the quartz crystal microbalance (QCM) mechanism and film bulk acoustic wave (FBAR) sensors, as well as SAW biosensors. The basic principle of operation of each mechanism, piezoelectric materials utilized with their desirable properties, challenges, fabrication methods, and typical biosensor applications in the past decade are also accordingly discussed. Section 3 gives a brief introduction to the use of BAW and SAW in microfluidic actuation and their application in the biosensing field.

Even though several review publications have reported pertinent acoustic wave-based biosensors and microfluidic actuation applications, they have concentrated on either SAW-based biosensors or actuation applications alone. Herein, we aim to bridge the gap between the entire realm of acoustic wave modes in biosensing, actuation applications, and their fundamental physics. An insight into the piezoelectric material types used for acoustic wave generation in the bioapplication area has also been studied with their relevant material properties. The general scope of this review is to furnish a coverage on all the acoustic wave modes that can be utilized for biosensing and actuation with fundamental operating principles, and the developments made in terms of their applications in biosensing in the past decade. The review paper also calls attention to the applications of these techniques in sensing bioanalytes, such as proteins, disease biomarkers, DNA, and pathogenic microorganisms, and discusses the challenges towards commercialization. Future perspectives of the integration of acoustic wave-based biosensors and microfluidic actuation on a single platform enable the realization of monolithic integrated real-time, user-friendly, low-cost, miniature LOC and POC devices. More challenges are yet to be overcome towards the integration of acoustic biosensors and microfluidic actuation in a single piezoelectric substrate, which needs a careful choice of acoustic modes to operate without signal attenuation. Studies in the area of thin-film piezoelectric material deposition methods, as well as film property characterizations, are also equally important and need to be reviewed towards the development of miniature acoustic wave-based biosensing platforms, which is out of the scope of this review.

## 2. Acoustic Biosensors

Acoustic biosensors play an important role in the development of miniaturized, portable devices for healthcare, food, environmental, and medical-related areas for consumer applications [26,38]. Accordingly, piezoelectric acoustic biosensors use acoustic waves generated by piezoelectric materials to extract information about the analytes [26]. For bulk material substrates, quartz, lithium tantalate (LiTaO_3_), and lithium niobate (LiNbO_3_) are commonly used [39]. Deposited piezoelectric thin films, including lead zirconate titanate (PZT), aluminium nitride (AlN), and zinc oxide (ZnO), are also used, providing enhanced piezoelectric properties with the capability of the miniaturization of acoustic devices [1].

An acoustic biosensor consists of mainly two components: a bioreceptor (sensing element) attached to an acoustic transducer (generates and detects acoustic waves). The bioreceptor interaction with the analyte results in changes in the density, viscosity, elasticity, or electrical properties of the sensing element. These changes affect the acoustic wave properties, such as wave velocity/phase shift, amplitude, and frequency, which are converted to an electrical output by the acoustic transducer [26]. A simplified diagram of this working principle is shown in Figure 2.

The acoustic waves are generated by the electromechanical transduction in the piezoelectric materials. The control of this mechanism is performed through the direct application of an alternating current (AC) electric field on a piezoelectric material that results in the generation of mechanical displacement by the inverse piezoelectric effect, which then propagates through the piezoelectric material as acoustic waves [40,41]. This property helps in the formation of compact and integrated piezoelectric acoustic biosensors by simply coating the bioreceptor layer onto the piezoelectric crystal [42].

For a homogenous non-piezoelectric solid, the application of mechanical stress results in elastic deformation or strain, which follows Hooke’s law. The wave equation for corresponding the elastic waves propagating in a homogenous non-piezoelectric solid can be represented as [41,43]
(1)$$\rho \frac{\partial^{2}u_{j}}{\partial t^{2}}{=c}_{\mathrm{ijkl}}\frac{\partial^{2}u_{k}}{\partial x_{i}x_{l}},$$

For a piezoelectric solid, the application of mechanical stress results in the development of an electric field along with the mechanical strain by the direct piezoelectric effect [44]. The constitutive equation for Hooke’s law for a piezoelectric material is given as [40,43]
(2)$$T_{\mathrm{ij}}{=c}_{\mathrm{ijkl}}S_{\mathrm{kl}}-e_{\mathrm{ijm}}E_{m},$$
(3)$$D_{i}{=e}_{\mathrm{ikl}}S_{\mathrm{kl}}+{ℇ}_{\mathrm{ik}}E_{k},$$
where *i*, *j*, *k*, *l*, and *m* represent the Cartesian coordinate system components with each taking values 1, 2, or 3. $x_{i}$ is the *i*^th^ position coordinate; *t* is the time, *ρ* represents the density of the solid; $u_{j}$ and $u_{k}$ are the mechanical particle displacement components in the *j* and *k* direction; $c_{\mathrm{ijkl}}$ represents the elastic modulus of the solid; $T_{\mathrm{ij}}$ represents the mechanical stress tensor component; $S_{\mathrm{kl}}$ represents the mechanical strain tensor component; $E_{m}$ is the electric field; *D_i_* is the electric displacement; ${ℇ}_{\mathrm{ik}}$ is the dielectric permittivity matrix; and $e_{\mathrm{ijm}}$ represents the piezoelectric coefficient tensor components. Thus, the generated acoustic waves in a piezoelectric solid have both the electric field and mechanical displacement associated with them. This renders uniqueness to the acoustic biosensors; they can detect both mechanical and electrical characteristic changes related to bioreceptor–target interactions [38]. The generated acoustic wave properties accordingly depend on the elastic, piezoelectric, and dielectric properties, as shown in Equations (2) and (3), which change based on the influence of the target analyte on the bioreceptors [43].

The bioreceptors used are biologically active materials, such as antibodies, oligonucleotides, or enzymes, that have a high selectivity to the analyte being detected [42,45]. They detect the target analytes and induce a change in the mass of the layer, which is detected by the acoustic wave, making the acoustic biosensors function as a gravimetric sensor [46,47]. Variation in the viscosity and viscoelastic properties of the adsorbed layer and the medium in contact with the acoustic waves can also be detected, which helps in the analysis of blood coagulation studies, cell adhesion, and complex biomolecules, such as nucleic acids [48,49,50,51]. Apart from being capable of detecting mass, the electroacoustic property of piezoelectric acoustic waves makes them sensitive to the dielectric properties, such as the conductivity and permittivity of the bioreceptor layer, or the medium in contact with it. This enables the acoustic biosensors to detect various complex biological samples, such as glucose, intracellular pH, and uric acid, in the human serum [15,18,52]. The enzymatic reaction between targets, such as glucose or uric acid, and the specific bioreceptor results in the changes in the pH and, hence, the conductivity of the medium in contact with the acoustic sensor. This induces an electro–acoustic interaction between the propagating acoustic waves and the surface charges in the medium, which affects the wave velocity and, hence, the frequency [53]. Acoustic wave biosensors, in general, can be used to detect a wide range of analytes varying from proteins, disease-causing biomarkers in humans and animals, and biomolecules, such as pesticides, toxins, complex nucleic acid molecules, and disease-causing pathogens [54,55,56,57]. The most common mode of operation of an acoustic wave device is as a resonator, in which the acoustic wave frequency becomes the detectable signal corresponding to the sensing of the analyte. The acoustic wave amplitude and phase shift can also be used as a detection signal for acoustic biosensors.

Piezoelectric material properties play an important role in the performance of acoustic biosensors. The electromechanical coupling coefficient (K^2^), acoustic wave velocity in the material, and temperature coefficient of frequency (TCF) are some of the parameters that affect the acoustic wave generation and, hence, the sensor performance. These properties will be discussed in detail in each of the BAW and SAW sensor categories, together with the typical piezoelectric materials used for each type. The performance of acoustic biosensors is characterized by measuring the mass sensitivity and limit of detection (LOD). Mass sensitivity refers to the change in the resonant frequency of acoustic waves caused by the mass added to the sensor surface. LOD refers to the minimum amount of mass that can be detected by the biosensor. Higher sensitivity and lower LOD are the required criteria for any biosensor, which enables the detection of smaller molecular weight analytes even at lower concentrations. Another important factor that determines the performance of biosensing in a liquid medium is the quality factor (Q factor). Q factor determines how sharp the resonance peak of the sensor is and helps to improve the sensitivity and dynamic range of the sensor [26]. Detailed explanations and calculations will be discussed in the following sections of the BAW and SAW biosensors.

Biosensing is generally performed in a liquid medium due to the properties of target analytes, such as biomolecules and cells [41]. This makes the properties and modes of acoustic waves crucial in choosing the appropriate type of acoustic biosensors for operating in a liquid medium [26]. Moreover, we introduce, here, the basic working principles of BAW- and SAW-based biosensors, the acoustic wave modes suitable for biosensing, together with the type of materials used, fabrication steps, and challenges, accordingly. The major applications of these biosensors in detecting target analytes, such as proteins, disease biomarkers, DNA, cells, and pathogenic microorganisms reported in the past decade, will also be discussed.

### 2.1. Bulk Acoustic Wave Biosensors

Bulk acoustic waves (BAWs) are generated when bulk piezoelectric elements, such as ceramic substrates or plates and thin films, are excited by metallic electrodes, generally Au coated on the top and bottom faces [26]. Upon the application of an alternating electric field (EF) across the electrodes, the potential difference induces a mechanical deformation due to the inverse piezoelectric effect mentioned earlier. This deformation, in turn, induces mechanical waves propagating into the bulk of the material perpendicular to its surface, hence its name. The wave travels through the material and is reflected from the bottom substrate surface, resulting in a constantly oscillating mechanical standing wave. The frequency of oscillation is characterized by the piezoelectric material density, elastic constant, and the thickness of the material [58]. BAWs can mainly propagate in either a longitudinal mode or transverse mode, based on the direction of particle displacement relative to the wave propagation direction. The propagation mode is mainly governed by the crystal cut orientation of the bulk piezoelectric material, as well as the orientation of electrodes with respect to the c-axis of piezoelectric thin films [41,59,60]. Transverse mode, also named thickness shear mode (TSM), has the particle displacement perpendicular to the wave propagation direction and parallel to the substrate surface. The longitudinal mode has particle displacement in the same direction as that of the wave propagation. The propagation modes are shown in Figure 3. For biosensing applications, the thickness shear mode of operation of BAWs is preferred, due to the reduced coupling of the acoustic energy with the liquid medium in contact, which results in reduced acoustic damping [61].

BAW biosensors are generally made up of two types of waveguiding structures, namely the bulk substrate and plate/thin-film membrane, as shown in Figure 4. Quartz crystal microbalance (QCM) falls under the bulk substrate category, in which the acoustic wave propagates into the bulk substrate. Film bulk acoustic wave resonators (FBARs) fall under the plate/thin-film category, in which the acoustic wave propagates in thin piezoelectric plates or film membranes [1,26]. Among the types of BAW biosensors mentioned above, the architecture for QCM and FBAR devices is similar, except for the use of the bulk substrate for the former and the thin piezoelectric deposited film membrane for the latter.

#### 2.1.1. Quartz Crystal Microbalance (QCM)

QCM is a BAW device that operates in the thickness shear mode (TSM) and it is also referred to as a TSM resonator. As the name implies, QCM has been widely used as an analytical gravimetric sensor for mass measurement [41]. The device architecture involves a quartz crystal of thickness *t_q_* with the top and bottom surfaces metalized with a circular electrode pattern of radius R, as shown in Figure 5a [41]. The top and the bottom surfaces of the quartz crystal deform in parallel but in the opposite direction, producing mechanical waves that travel through the bulk when an alternating EF is applied [58]. The particle displacement profile in the QCM is shown in Figure 5b. The acoustic waves traveling in the bulk are reflected from the substrate surfaces, forming standing waves of oscillations in the bulk when constructive interference occurs. This condition requires the thickness of the quartz crystal to be equal to an integer multiple of half the acoustic wavelength $(\frac{n\lambda }{2}$). The fundamental resonant frequency (*f_0_*) of the oscillations in the QCM is given by [58]
(4)$$f_{0}=\frac{v_{p}}{{2t}_{q}}=\frac{1}{{2t}_{q}}\sqrt{\frac{\mu_{q}}{\rho_{q}}},$$
where $v_{p}$ is the phase velocity of the acoustic wave, $\mu_{q}$ is the elastic constant, and $\rho_{q}$ is the density of the quartz crystal. Equation (4) shows that the resonant frequency of QCM depends on the thickness of the crystal and can provide information about material-specific parameters. The typical operating frequency range of the QCM is 5 MHz to 30 MHz [62]. Due to the shear deformation of the particles in the QCM, they are suitable for biosensing in a liquid medium with reduced damping [63].

The complete description of QCM oscillations can be understood from its electrical equivalent circuit. In the case of the no loading condition, the equivalent circuit can be modeled using lumped elements called the Butterworth–Van–Dyke (BVD) circuit, as shown in Figure 5c [41]. This circuit helps to establish the relation of the circuit elements to the QCM properties, as well as to the adsorbed mass and the liquid in contact with the QCM when operated as a sensor [68]. The motional resistance R_1_ in the series branch of the circuit, represents the energy dissipation due to multiple factors, such as the viscous effects, damping, and internal friction. The resonant frequency of the QCM can be obtained from the impedance response curve at the frequency corresponding to the 0 phase shift. In the absence of damping (R_1_ = 0), two resonant frequencies are corresponding to the minimum and maximum impedance at 0 phase shift, named as resonant frequency f_r_ and anti-resonant frequency f_a_, respectively, as shown in Figure 5d. With damping (R_1_ ≠ 0), the separation between the resonant frequencies increases due to changes in the phase shift induced by the energy dissipation, as shown in Figure 5e. This paves way for new possibilities in the QCM for biosensor applications, such as viscosity sensing or even identifying the single point mutations in the DNA, or distinguishing DNA molecules with a similar mass but with different structures [69]. The measurement of the impedance response can be performed using a network analyzer or a frequency counter [41,70].

##### 2.1.1.1. QCM as Biosensors: Operating Principle

The basic architecture of a QCM as a biosensor involves the functionalization of the top Au electrode with a bioreceptor layer, as shown in Figure 6a. Only one of the QCM electrodes faces the liquid biosensing medium, while the other electrode is isolated against air. This is performed to avoid the risk of electrical coupling between the two electrodes when operating in a low electrical impedance medium [23]. Bioreceptors can be a layer of antigen, antibody, aptamers, nucleic acids, synthetic peptide chains, or molecularly imprinted polymers (MIP), which are bound to the Au electrode by physisorption, covalent bonding, or electrostatic interaction [71,72,73,74,75]. The bioreceptors make the QCM sensors selective and specific, which selectively adsorb the target analyte from the sensing medium and enables label-free detection.

The use of QCM as an analytical device gained significance after Sauerbrey discovered that a linear relationship existed between the frequency response and mass added onto the QCM device [76]. The basic principle behind the application of QCM as a sensing element is gravimetric sensing. The addition of mass onto the surface electrode of the quartz crystal in the form of thin, rigid films, is equivalent to an increase in the thickness of the quartz crystal, which can lead to a decrease in the resonant frequency, as shown in Figure 6a. Hence, the electrical quantity that responds to the analyte binding onto the bioreceptor is the frequency. The change in frequency Δf resulting from the addition of the mass of Δm is given by the Sauerbrey equation [76,77]
(5)$$\Delta f=-\frac{{2f}_{0}^{2}}{A\sqrt{\mu_{q}\rho_{q}}}\Delta m,$$
(6)$$S_{m}=\underset{\Delta m\to 0}{\lim}\left(\frac{\Delta f}{f_{0}}\right)\left(\frac{1}{\Delta m}\right),$$
where *A* is the area of the sensor surface and $S_{m}$ represents the mass sensitivity of the sensor. $S_{m}$ is used to characterize the sensor performance that describes the change in resonant frequency with the added mass, where *Δm* is the mass added to the sensor per unit area. By combining Equations (4) and (5), a general expression for all resonant mass sensors can be obtained as [77]
(7)$$\frac{\Delta f}{f_{0}}=-\frac{\rho_{m}t_{m}}{\rho_{q}t_{q}}=-\frac{\Delta m}{m_{0}},$$
where $\rho_{m}$ and $t_{m}$ are the density and thickness of the added mass and $\rho_{q}$ and $t_{q}$ are the density and thickness of the resonator layer, respectively. Equation (5) shows that the sensitivity of the sensor has a quadratic relationship with the resonant frequency. The linear mass–frequency change relationship, as shown in Equation (5), is applicable only if the sensing occurs in air or vacuum, and the adsorbed film has the same property as the quartz crystal and is uniform, rigid, and thin. Another parameter that determines the performance of the gravimetric sensor is the quality (Q) factor. The Q factor can be defined as the ratio of the energy stored in the QCM resonator to the energy dissipated. It is measured as the ratio of the resonant frequency to the full-width half-maximum of the impedance response. The electrode thickness and material play an important role in the energy trapped in the QCM resonator and, hence, play an important role in improving the Q factor [41].

Equation (5) may not be sufficient to describe the frequency shift produced in the application of biosensing in the liquid medium. The viscoelastic properties of the medium also influence the QCM performance. When one of the surfaces of the QCM sensor is exposed to a viscous liquid, the shear acoustic waves extend into the liquid medium as a highly damped sinusoidal wave of wave vector k, angular frequency ω with a penetration depth δ, as shown in Figure 6b. The performance of the QCM in the viscous liquid medium has been explained by Kanazawa and Gordon, where the resonant frequency ($f_{0})$ of the resonator decreases proportional to the viscosity ($\eta_{l})$ and density ($\rho_{L})$ of the liquid in contact, and is given by [78]
(8)$$\Delta f=-f_{0}{}^{\frac{3}{2}}\sqrt{\frac{\eta_{l}\rho_{L}}{{\Pi \mu }_{q}\rho_{q}}},$$
(9)$$\delta =\frac{1}{k}=\sqrt{\frac{{2\eta }_{l}}{{\omega \rho }_{L}}},$$

Equations (5) and (8) indicate that the QCM will respond to both mass and liquid loading, and that the total frequency change will have a contribution from both [68]. When the QCM is in contact with a liquid medium, acoustic energy loss and dissipation occur and this can lead to the lowering of the Q factor, as shown in Figure 6c. The lowering of the Q factor due to liquid loading can be explained using the BVD model, as shown in Figure 6d, with loading impedance Z^1^_m_ added to the series motional arm. The energy dissipation, due to the liquid loading, results in an increase in motional impedance and causes the broadening and reduction in amplitude of the impedance response [58,68].

Even though the operating frequency range of the QCM biosensor is lower, leading to a lower sensitivity compared to SAW biosensors, it is being widely applied to biosensing. This is due to the ease of fabrication, low cost, excellent chemical and thermal stability of the quartz crystal, biocompatibility, and ease of measurement data analysis [58]. The target analyte bound onto the bioreceptor on the QCM sensor surface can act as a rigid thin film, which leads to frequency reduction as per the Sauerbrey equation, due to the added mass. The bound target can also act as a viscoelastic film, especially in the case of cells, complex biomolecules, and nucleic acids attached to the sensor surface. This requires a careful frequency response estimation to differentiate the frequency variation contribution from the adsorbed mass and viscoelastic property.

The performance of the QCM biosensor is normally quantified using mass sensitivity (S_m_), as represented by Equation (6), and limit of detection (LOD) [77]. LOD represents the minimum detectable mass by the sensor and can be calculated from the response of the QCM to the baseline buffer solution without the target analyte. LOD is calculated as [80]
(10)LOD=Δf+3 ×SD,
where Δf is the mean response and *SD* is the standard deviation of the baseline response. This method is named blank determination. The LOD can also be calculated from the linear calibration curve, where the sensor response is linearly related to the target concentration within a limited range as LOD =$\frac{3\sigma }{S_{f}}$. σ represents the standard deviation of the response and $S_{f}$ represents the sensitivity, which is the slope of the sensor response. This method is named as the linear regression method [80].

##### 2.1.1.2. Challenges in QCM Biosensing

For QCM biosensing, the complex buffer medium and the biomolecules adsorbed onto the QCM itself can have a different viscoelastic property when compared to the quartz crystal. Parts of the elastic adsorbed biomolecular layer follow quartz crystal oscillations, whereas the viscous portion resists moving along with the oscillations, leading to the energy dissipation into the medium through viscous coupling. This results in the damping of the oscillation frequency, compounded with the frequency reduction due to the adsorbed mass [41,81]. The resonant frequency of the QCM is also affected by the surface roughness of the crystal; the roughness of the adsorbed biomolecule layer; the hydrophobicity of the surface; the properties of the buffer medium, such as the conductivity, permittivity, and ionic strength; the presence of the surface charge; and the electrochemical double layer [82,83,84,85,86,87]. The effect of all these interferences gives rise to an overestimation of the mass attached to the sensor, if the Sauerbrey equation alone is utilized for analysis.

One method to subtract the interfering effects on the resonant frequency, is to use the reference channel measurement along with the active sensor or by using a QCM array [88,89,90]. The usage of QCM array with multiple Au electrodes within the same quartz substrate has the advantage of frequency and temperature stability, compared to using a separate reference QCM. Another method to decode the contribution to the resonance frequency variation, by factors other than the adsorbed mass, is impedance analysis using a network analyzer. This method provides a way to estimate both the resonant frequency variation and the damping induced by the various external and interfering factors [67]. A third method, is the use of the QCM-D mode, where D is the dissipation factor that provides information about the frequency damping caused by the non-mass added effects. Two pieces of information can be derived from the QCM oscillator circuit in this mode: the temporal variation of the resonant frequency and dissipation factor D, which is the inverse of the Q factor [91,92].

Apart from these challenges, there is a limit to the sensitivity and, hence, the minimum detectable mass that can be achieved using the QCM [46]. The lower sensitivity of the QCM arises from the increased base mass of the thick piezoelectric crystal being used, resulting in a lower resonant frequency. It is not possible to reduce the thickness of the quartz crystal beyond a limit, which would make the device fragile. This puts a limit on the resonant frequency of the QCM of up to 30 MHz and, hence, affects the sensitivity, as shown by Equation (5). A method that is used to improve the frequency change for low molecular weight targets and low concentration analytes is the signal amplification technique [93]. Signal amplification strategies tend to enhance the mass change sensitivity by adding extra mass onto the detected target in the form of nanoclusters. The added mass further lowers the resonant frequency that becomes measurable. Nanoparticles (NP), such as AuNP, AgNP, SiNP, and TiO_2,_ and magnetic nanobeads are conjugated with a detection antibody [94,95,96,97,98]. This conjugate is bound to the captured target on the QCM forming a sandwich assay, which helps to enhance the frequency drift. A derivative of the QCM sensor, termed as the high fundamental frequency (HFF) QCM, with the quartz substrate being thinned at the center electrode area, has also been reported for a higher resonant frequency operation with improved sensitivity [99]. QCM devices are also not suitable for miniaturization, integration of electronic readouts, and microfluidic devices as they utilize the full thickness of the quartz substrate used [100].

##### 2.1.1.3. Piezoelectric Materials and Fabrication

AT-cut quartz is the most commonly used piezoelectric material for the QCM. It is obtained by cutting the quartz crystal at an angle of 35.25° to the optical axis [41]. AT represents the temperature stabilized cut, whereby the temperature coefficients of the lattice will have minimal impact on the crystal performance. This specific crystal cut is used due to its excellent chemical, mechanical, and thermal stability. The cut of the quartz crystal determines the range of the operating frequency and the propagation modes supported due to the anisotropy of piezoelectric materials. Apart from these properties, AT-cut quartz crystals have a near-zero temperature coefficient of frequency (TCF), which makes the frequency response of the QCM stable against temperature variations. This makes them excellent for biosensing applications [101,102].

The fabrication process for the QCM for biosensors is quite simple, with circular polished quartz crystal being deposited with an Au electrode of a thickness of around 50 nM–100 nM on both surfaces, as shown in Figure 7a. The typical diameter of the quartz crystal is in the range of about 12 mm to 14 mm and the Au electrode is about 4 mm to 6 mm to overcome the high demands on the QCM imposed by the liquid loading in biosensing [103]. For operating in a liquid medium for biosensing, the QCM resonators can be integrated with a flow-through cell with one side of the QCM sealed using a rubber casing to make sure that only one sensor surface is in contact with the sensing medium [41]. Another method of operation is the dip and dry method, for which the resonator is not in contact with the liquid while the measurement is taken [41,104,105]. This method does not allow for real-time measurements and has an added risk of contamination during the drying phase. A QCM sensor with an integrated PDMS flow channel for sensing in liquid medium is shown in Figure 7b.

A class of QCM resonators, referred to as high fundamental frequency (HFF) QCM, has been designed to operate at a higher frequency range of up to 50 MHz to improve the sensitivity [99]. The selected area near the circular electrode at the center of the quartz crystal is thinned down using standard photolithography and etching. The remaining portion of the quartz crystal is untouched, so that better mechanical stability can be achieved. The fabricated HFF QCM device is shown in Figure 7c, in which the center portion of the quartz with a diameter of 3 mm is thinned down. A fundamental frequency as high as 200 MHz can be achieved using this method. Both wet etching and deep reactive ion etching (DRIE) can be used for the removal of quartz [106,107].

##### 2.1.1.4. Applications of QCM in Biosensing

QCM has been widely used in biosensing, due to the ease of manufacturing, low cost, and capability of providing a rapid, real-time, simple, and label-free technique for sensing a wide range of target analytes. Conventional methods for the detection of biological analytes, such as enzyme-linked immunosorbent assay (ELISA) and polymerase chain reaction (PCR), are highly sensitive and specific, but the results are not obtained in real-time and need complex sample pre-treatments [108]. QCM biosensors, on the other hand, are easy to operate and do not require highly skilled expertise, making them attractive for POC applications. Moreover, the sensitivity of the QCM to the viscoelastic properties of the adsorbed layer, along with the mass loading, makes them suitable for studying complex biomolecules such as nucleic acid, and cell structures and their morphology. QCM-D mode of the biosensor enables the monitoring of acoustic energy dissipation via monitoring the dissipation factor and provides an added advantage for biosensing.

Various bioreceptors are used in QCM biosensors, and their selection depends on the type of target analyte. The commonly used bioreceptors are the antibodies specific to the antigens being detected; synthetic oligonucleotides, named as the aptamers specific to various proteins and small biomolecules; molecularly imprinted polymers (MIP); and oligonucleotide single-stranded DNA (ssDNA) probes [73,75,98,109]. Depending on the bioreceptor–target interaction, biosensors can be classified as immunosensors (based on the antibody–antigen interaction), aptasensors (where aptamers are used as bioreceptors), and DNA biosensors (where oligonucleotide probes are bioreceptors). They can also be grouped based on the target analytes detected. QCM biosensors are used to detect a variety of targets, such as the proteins and biomolecules in human serum, which act as disease biomarkers [110,111,112]. They are used to study the affinity interactions between the biomolecules, cell detection, and separation, with studies including cell adhesion and the interaction with the substrates [113,114,115,116]. Various complex DNA molecules and pathogenic microorganisms, including foodborne and waterborne pathogens, disease-causing viruses, and bacteriophages in humans, plants, and animals, can also be detected using QCM as a biosensor [73,95,117,118]. The following section focuses on the applications of QCM biosensors, in terms of the above-mentioned targets being detected with their operating principles and performance. Appendix A in Appendix A provides a comprehensive list of various QCM biosensing applications for the detection of various types of analytes reported in the past decade.

(i)Proteins and Biomolecular Detection

Proteins and biomolecules in human serum play an important role in the diagnosis of various diseases and act as biomarkers for early detection [119]. The QCM biosensor with AT-cut quartz has been used for the detection of various disease biomarkers found in human serum, namely the carcinoembryonic antigen (CEA); prostate-specific antigen (PSA); CD63 positive exosomes; tuberculosis biomarker; Dengue fever biomarker (non-structural protein-1 (NS1)); pancreatic disease biomarker (trypsin); hepatitis B core antigen (HBcAg); protein biomarker for malaria (PfHRP-2); and human immunodeficiency virus (HIV) biomarker (HIV-1 glycoprotein 41 (gp41), HIV-1 p24 antigen) [75,97,98,111,112,120,121,122,123,124,125,126,127,128]. Apart from the disease biomarkers, QCM has also been used for the detection of various proteins and amino acids in human serum, such as tryptophan (Trp), immunoglobulin (IgG), hemoglobin, and plasmin (PLA), which act as early disease diagnosis markers [123,129,130,131,132]. QCM has also been used for the study of the affinity interaction of protein biomolecules [133,134]. The bioreceptors used are antibodies specific to the target, MIP, short amino acid peptide chains (Pcc), or protein layers for assessing the protease activity.

CEA is one of the most commonly used cancer biomarkers for the clinical diagnosis of lung and breast cancer. A QCM sensor operating at 5 MHz was reported for the real-time detection of CEA using an anti-CEA monoclonal antibody as a bioreceptor [112,135]. A novel method of a graphene oxide-Au nanoparticle-coated Au electrode was used for the immobilization of the anti-CEA antibody bioreceptor [112]. The schematic representation of this direct immunoassay is shown in Figure 8a. The reduction in the resonant frequency of the QCM was observed with the target CEA captured by the antibody due to mass loading. The sensor responded to varying concentrations of the target CEA with an increase in the frequency shift, as shown in Figure 8b. The device showed a LOD of 0.06 ng/mL for the QCM with an oxygen plasma-treated Au electrode. A more sensitive and lower LOD for the CEA detection was demonstrated with the QCM combined with enzymatic biocatalytic precipitation (EBCP) for the mass amplification [135]. The technique employed here further lowered the LOD of the QCM biosensor to 7.8 pg/mL.

The protease activity of some of the target protein species, such as trypsin or plasmin, can be utilized as the detection principle using the QCM [75,122,132,136]. The bioreceptors used for the detection are either an artificially synthesized peptide chain (Pcc) or proteins, such as ß-casein, instead of antibodies. The target protein species cleave the bioreceptor bonds leading to a mass reduction that can be detected using the frequency increase in the QCM. A synthetic peptide-based QCM sensor with AT-cut quartz operating at 7.995 MHz was reported for the detection of trypsin (biomarker for pancreatitis) and exhibited an LOD of 8.6 ng/mL [75]. A similar protease activity-based detection principle was used for sensing proteolysis in milk, involving plasmin (PLA) using the QCM [131,132,136,137].

A QCM with a molecularly imprinted polymer (MIP) as a bioreceptor layer on the Au electrode has been reported for the detection of proteins and disease biomarkers in human serum, such as tryptophan (Trp) and the HIV-1 glycoprotein [124,129]. MIPs are a special type of polymer, synthesized with a cavity that specifically matches the shape, size, and functional groups of the target analytes [129]. L-Tryptophan (Trp), which is one of the essential amino acids in the human body for the development of neurotransmitters, is considered a biomarker for neurological disfunctions [138]. The MIP synthesized using methacrylic acid (MAA) as a monomer with a Trp template and an ethylene glycol dimethacrylate (EGDMA) crosslinker was coated onto the Au electrode of QCM, to act as a bioreceptor for the specific and sensitive detection of Trp [129]. The device showed an LOD of 0.73 ng/mL. The schematic of the QCM sensor with the MIP as the bioreceptor, is shown in Figure 9.

(ii)Cell adhesion and cell detection

The QCM’s ability to detect both viscoelastic and mass loading by the measurement of the motional resistance and resonant frequency, makes them a good candidate for disease-causing cell detection, such as Leukemia cells, human breast cancer cells, and the metastatic potential of cancer cells [114,116,139,140,141]. Antibodies, or synthetic aptamers specific to cell membrane proteins are used as bioreceptors to initiate the detection of cells. QCM based biosensors, have also been developed for the real-time, label-free, and rapid detection of the metastatic potential of human breast cancer cells. The metastatic potential of cancer cells plays an important role in the spreading of tumors from the original location to secondary organs [142]. Sensing is based on the detection of certain molecular changes in the cancer cell membrane, which acts as a signal for metastatic potential. They are manipulated as the overexpression of adhesion molecules, such as CD44, Notch 4 receptor, folate receptor, or HER/neu, to name a few, on the tumor cell wall membrane. These molecules serve as the target for the QCM biosensors [114,116,140,141]. Antibodies specific to these molecular expressions are immobilized on the Au electrode, which serves to detect the metastatic tumor cells and causes a reduction in the QCM resonant frequency.

The QCM Au electrode surface coated with Poly (2-hydroxyethyl methacrylate) nanoparticles (PHEMA-NPs), were used for the immobilization of antibodies specific to the notch-4 receptor for the detection of the metastatic potential of human breast cancer cells with an overexpression of the Notch 4 receptor [114,141]. The schematic of the QCM chip used for the detection and resulting frequency response, is shown in Figure 10. The device exhibited excellent selectivity with an LOD of 12 cells/mL and was capable of reuse after the regeneration of the sensor surface. The metastatic potential of the tumor cell can also be detected from the cell mechanical property, including stiffness [143]. Metastatic tumor cells tend to be less stiff, which can be detected by the QCM-D via the monitoring of the dissipation factor. The QCM-D, with the Au electrode coated with poly-L-lysine, was used for the capture of CD44 overexpressed tumor cells [141].

QCM has also been widely used for the non-invasive and real-time study of the cell–substrate adhesion, related to hemostasis, the morphological study of cells, and cell motility, which finds application in cell biology and medical research [17,113,144,145,146]. A detailed review of the QCM in cell-substrate adhesion can be found in this reference and will not be covered in this review [113].

A QCM operating at a 5 MHz frequency was used for the real-time, label-free study of kinetics involved in primary hemostasis [17,144]. The QCM Au electrode was functionalized with collagen to promote cell adhesion [147]. This initiated a platelet response from the blood to mimic hemostasis with the device, as shown in Figure 11a. The sensor was integrated with Poly (methyl methacrylate) (PMMA) fluidic cells and shear rates were controlled to imitate the blood microcirculation rate in the human body. The AFM height and deflection trace of the platelet from the blood adhered to the collagen-coated QCM surface, is shown in Figure 11b. the detected drop in the sensor frequency was due to the initial addition of blood, and. Subsequently. due to the viscoelastic and mass loading induced by the platelet adhesion to the collagen-coated sensor surface, as shown in Figure 11c.

The adhesion property of the cells onto the QCM substrate surface led to the realization of cell-based sensors. The cells that adhered to the QCM Au electrode served as bioreceptors. Cell-based sensors work on the principle of the response of the adhered cells on the QCM to the target. This response originates from the morphological changes of the cells initiated by the interaction with the target species. The morphological changes can include cell death or inactivity, which can be detected via the QCM frequency response through the mass loading effect [115,148,149,150]. This method has been used to study glycosylation at the cell surface [115,150]. The QCM-D cell-based biosensor has also been reported to differentiate diabetic red blood cells from healthy cells, based on the specific interaction of the diabetic cell with the adhered endothelial cells on the QCM substrate acting as a bioreceptor [149].

(iii)DNA Biosensors

Nucleic acid biosensors have been receiving importance as a clinical diagnostic tool, due to the significance of gene study and infectious disease analysis [20]. An accurate DNA biosensor, with rapid, real-time, and label-free detection, suitable for POC applications, can be realized by using QCM for detecting DNA gene sequences. By using a complementary oligonucleotide probe as a bioreceptor on the QCM, the hybridization of target DNA results in the frequency drop of the biosensor due to mass addition. Improving the detection sensitivity of QCM is achieved by using mass amplification methods to enhance the frequency drop of the QCM [95,96,151].

MicroRNA is a single-stranded small-size nucleotide, which is considered as a disease biomarker for cancer and immune disorders [152]. A sensitive and label-free nucleic biosensor for the detection of microRNA-21 (miR-21) has been reported using a QCM operating at a 9 MHz resonant frequency with an LOD of 0.87 pM [96]. Signal amplification through mass enhancement was achieved by introducing the TiO_2_ nano particle-detection ssDNA probe conjugate, which hybridized with the captured target miR-21 on the QCM Au surface. An AgNO_3_ solution was added to the sensor surface along with exposure to UV light, leading to the photocatalytic deposition of Ag onto the TiO_2_ nanoparticles for further mass amplification.

Detecting gene mutations in humans and animals plays an important role in clinical research, which determines the response to disease-causing microorganisms as well as to drugs [153]. DNA biosensors using QCM are also used for studying gene mutations and single nucleotide polymorphism (SNP), in which the DNA sequences differ in a single nucleotide at a specific location [108,151,154,155]. QCM with a resonant frequency of 5 MHz, using modified nanoporous electrodes, was reported for the label-free, sensitive, in situ detection of epidermal growth factor receptor (EGFR) gene mutation, which is a transmembrane protein [108]. Detecting EGFR mutation plays an important role in determining the efficacy of lung cancer treatment drugs [110]. The Au electrode of QCM was modified using the electrochemical deposition of Ag and Au alloys, to form a nanoporous structure. This offered more surface area for the immobilization of the complementary ssDNA probe specific to the EGFR mutant gene sequence. The sensor showed an LOD of 1 nM and exhibited excellent specificity to the EGFR mutant [108].

A reusable, sensitive, and specific QCM DNA biosensor, for the detection of single-base gene mutation, was developed on QCM by strand displacement reaction (SDR) with an LOD of 0.8 nM [155]. The displacement was initiated at the complementary overhang single-stranded portion of the prehybridized duplex, which is called the toe hold domain. A unique mechanism for the successive detection of gene mutation and surface regeneration of the QCM was demonstrated, by utilizing the SDR involving a capture probe and a reporter probe, as shown in Figure 12. The toe hold domain was designed to be present at the overhang portion of the reporter DNA, to complement to the target mutant DNA. It initiated the SDR upon the detection of the target mutant DNA, leading to a sharp rise in the QCM resonant frequency due to the unloading of the reporter DNA by the SDR, as shown in the real-time frequency monitoring curve, in Figure 12. The detection of the target is accompanied by surface regeneration, due to the removal of the reporter probe.

DNA biosensors are also widely used for the label-free detection of viral infectious disease-causing microorganism gene sequences, including the hepatitis B virus (HBV), vaccinia virus, human papilloma virus (HPV), malaria virus, and tuberculosis, in humans, as well as disease-causing pathogens in animals [71,109,156,157,158,159,160,161]. DNA probes specific to the target virus gene sequences are used as the bioreceptors for sensing.

The rapid and label-free detection of HBV viruses enables QCM to be used for portable POC sensors [109,158]. Hepatitis B virus (HBV) detection, in a single step, was reported using a QCM-D resonator by immobilizing the Au electrode with an ssDNA probe, that selectively hybridized with the target HVB gene sequence [109]. To enhance the detection sensitivity of QCM to the HBV virus gene sequence, the in situ rolling circle amplification (RCA) of the detected target sequence on the QCM biosensor surface was reported, with an enhanced LOD of 10^4^ copies/mL [158]. The DNA sensor for the detection of *Ehrlichia canis* was reported, using a dual-electrode QCM resonator with a resonant frequency of 10 MHz [156]. The device schematic of the detection process and real-time frequency response is shown in Figure 13. The sensor exhibited selectivity to the target DNA sequence and an LOD of 22 copy numbers/mL of *E. canis*.

(iv)Pathogenic Immunosensor: Virus and Bacterial Microorganism Detection

QCM biosensors are used for the label-free, real-time, and rapid detection of disease-causing viruses and foodborne and waterborne bacteria. The direct detection of viruses, such as the avian influenza virus (AIV) (H5N1), and plant viruses, such as the maize chlorotic mottle virus, has been reported with the QCM biosensor [72,94,118,162,163]. They are designed with antibodies, aptamers specific to virus surface proteins, or MIP as bioreceptors to directly detect the target pathogens. Inactivated viral particles are commonly used for detection experiments.

The QCM biosensor for the label-free direct detection of the AIV H5N1 virus, which is the cause of highly pathogenic avian influenza, has been reported with various types of bioreceptors, such as aptamers and MIP [72,94,118,162]. They have been demonstrated for rapid detection of AIV in agricultural chicken swabs, which showed the capability of the device to be applied in clinical, food, and environmental samples [94,118]. A QCM resonator of 7.995 MHz was designed with a nanoporous Au film coating, for the sensitive detection of AIV H5N1 [118]. A synthetic aptamer specific to the H5N1 surface protein was used as the bioreceptor. The schematic of the surface functionalization on the nanoporous electrode is given in Figure 14a. The device reported an LOD of 2^−4^ HAU (hemagglutinating unit)/50 µL with a total assay time of 10 min, which is much faster compared to conventional PCR and ELISA tests. Other MIP polymers, with templates specific to the virus subtypes, were also used as bioreceptors for AIV subtype detection [72].

Various foodborne and waterborne pathogens, such as *Escherichia coli* O157:H7 (*E. coli*), *Salmonella*, and *Staphylococcus aureus*, are also detected using QCM [73,74,164,165,166,167]. QCM biosensors have been reported for the rapid and label-free detection of *Salmonella* in real food samples, with good sensitivity and selectivity by using antibodies or aptamers as bioreceptors [73,74,164]. The QCM biosensor with improved device sensitivity for the detection of *Salmonella*, was reported by the amplification method with the AuNP-detection antibody conjugate [164]. This formed a sandwich assay with an enhanced LOD of 10 colony-forming units/mL (CFU/mL) and a detection time of 12 min.

*E. coli* is another dangerous foodborne pathogen, whose rapid and label-free detection has been demonstrated with QCM using antibodies or aptamer bioreceptors [166,167]. An aptasensor with a QCM resonator with a resonant frequency of 7.995 MHz, was designed for the rapid and sensitive detection of *E. coli* O157:H7 [166]. A synthetic aptamer specific to *E. coli* was obtained by using the whole-bacterium SELEX (Systematic Evolution of Ligands by Exponential Enrichment) process, to act as a bioreceptor. The schematic of the aptasensor detection of target *E. coli* with frequency response, is shown in Figure 14b,c. The device exhibited excellent specificity to the target *E. coli*, in the presence of other interfering bacteria. An LOD of 1.46 × 10^3^ CFU/mL with a total assay time of 50 min was also observed, making it a rapid detection technique [166].

#### 2.1.2. Film Bulk Acoustic Resonator (FBAR) Biosensors

Even though QCMs have been widely used for biosensing, the typical lateral dimensions of the QCM are in the range of mm to cm, making them less sensitive towards micron-sized particles [77]. This becomes challenging in biosensing, especially in the area of single cells and molecules. The resonant frequency of QCM is also lower, which reduces the sensitivity. Moreover, the use of single-crystalline bulk substrates in QCM makes their bulk manufacturing, miniaturization, and integration of electronic circuits difficult. FBAR belongs to the BAW resonator category, with thin-film piezoelectric materials being used as an active medium for wave generation and propagation. The operating principle of FBAR devices is the same as that of the QCM, with acoustic waves traveling in the bulk of the piezoelectric thin films, such as AlN, ZnO, PZT, GaAs, and PVDF [100,168].

The basic architecture of FBAR devices includes a thin piezoelectric deposited film sandwiched between the top and bottom metal electrodes. The application of high-frequency alternating EF induces mechanical acoustic waves, which travel into the bulk of the piezo thin films. The resonant frequency of FBAR devices is given by the same Equation (4), as that of QCM, and depends on the thickness of the piezoelectric thin films. It is, typically, an order of magnitude higher compared to the QCM and SAW resonators. The resonant frequency of FBAR, ranges from lower GHz to a few tens of GHz [169]. The higher resonant frequency is due to the lower base mass of the FBAR devices and thinner piezoelectric deposited films, with thickness in the range of a few µm up to around 3 µm. The higher frequency, coupled with the better piezoelectric coefficient, d_33_, electromechanical coupling coefficient (K^2^), and increased acoustic velocity in thin films, such as AlN and ZnO, enables the design of FBAR biosensors with higher mass sensitivity and a better Q factor [100,170].

The typical impedance response of the FBAR is similar to that of QCM, as shown in Figure 5d. The Q factor represents the energy storing capability and damping of the resonator. The Q factor is given by [171]
(11)$$Q=\frac{f_{0}}{2}\left(\frac{\partial \emptyset_{z}}{\partial f_{0}}\right),$$
where $f_{0}$ is the resonant frequency, and $\emptyset_{z}$ is the phase of the electrical impedance of the FBAR resonator. FBAR devices fabricated by using Si micromachining techniques, have smaller dimensions within 1 mm^2^, can be manufactured in bulk quantities, and can be integrated with standard CMOS circuits for signal processing due to the CMOS compatibility of the sputtered piezoelectric thin films [49,172]. The active sensing area of the FBAR devices is smaller due to the miniature size, and leads to a reduced volume of sample consumption. They can also be integrated with microfluidic channels for sensing in the liquid medium, enabling them as a candidate for LOC applications [173].

Since the thin film deposition of piezoelectric materials needs substrate support, the acoustic energy leakage to the bulk is a concern. The operation of FBAR requires the isolation of the active acoustic wave generation area from the substrate, to minimize the acoustic energy leakage to the bulk [59]. There are two possible configurations of FBAR devices to achieve acoustic isolation and, hence, improve the Q factor. One is with the free-standing supporting membranes of SiO_2_ or Si_3_N_4_ coated with piezoelectric thin films formed by the backside cavity, as shown in Figure 15a. The design decouples the acoustic energy from the bulk Si substrate, by using the backside cavity and improves the acoustic energy trapping. This helps in the improvement of the Q factor and also enables the option of integrating the microfluidic channel beneath the membrane [174]. However, the fabrication step is complex and involves Si micromachining for the release of the membrane and affects the yield rate. The second configuration uses alternating layers of low and high acoustic impedance materials of thickness λ/4, called Bragg reflectors, which represent acoustic impedance mismatch. This helps to trap the acoustic energy without leaking to the bulk Si substrate and, hence, improves the Q factor. This design is referred to as solidly mounted resonators (SMR), as shown in Figure 15b. The structure is mechanically stable due to the absence of the backside cavity. However, the fabrication of SMR devices is costly and time-consuming due to the deposition of multiple layers in the Bragg reflectors. BAW propagation in FBAR devices can have different propagation modes, depending on the particle displacement relative to the wave propagation, as will be discussed in detail in the following section.

##### 2.1.2.1. Propagation Modes in FBAR

Depending on the crystal orientation of sputtered piezoelectric thin films and the electrode configuration, FBAR has, mainly, three propagation modes, as shown in Figure 16. The careful selection of the mode is required for biosensing in the liquid medium, taking into consideration the acoustic energy damping and Q factor reduction in the presence of the liquid medium.

(i)Thickness Extensional Mode (TE)

The TE mode involves the c-axis oriented piezoelectric thin films, which are perpendicular to the substrate surface with electrodes deposited on the top and bottom surface. The c-axis refers to the crystallographic c-axis of the piezoelectric crystal with either a perovskite or wurtzite structure. The applied electric field excites a longitudinal mode displacement in the thickness direction of the thin film. It gives the highest acoustic velocity with a high Q factor in the air [77]. However, this mode has limitations in biosensing due to the damping of acoustic energy in the liquid medium [175].

(ii)Thickness Shear Mode (TS)

The TS mode is excited in the piezoelectric thin films with a tilted c-axis, with respect to the substrate surface with top and bottom metal electrodes. The angle of tilt can be varied from 10° to 35° [46,173]. Tilted piezoelectric films support both shear mode and longitudinal mode, depending on the angle in degrees between the applied electric field and c-axis of the film. Pure shear mode occurs when the electric field and c-axis of the film become perpendicular. It can be achieved by laterally exciting the c-axis oriented piezoelectric film with coplanar electrodes on one surface [59,171,175]. The mode is termed as the lateral filed excitation (LFE) [176]. As shown in Figure 17, the shear mode has a better Q factor in liquid, when compared to the longitudinal mode, due to less coupling of the acoustic energy into the liquid medium in the shear mode. Schematic of TS mode excited device with tilted c-axis is shown in Figure 18a and LFE device with coplanar electrodes on c-axis oriented film is shown in Figure 18b.

(iii)Lateral Extensional Mode (LE)

The longitudinal mode in the lateral direction is termed the LE mode. The structural vibration occurs in a dilation pattern and can also be termed as the contour mode. The mode can be excited by patterning electrodes on one side of the thin piezoelectric membranes, as shown in Figure 18c [177]. This mode is suitable for biosensing, as the longitudinal wave contacts the liquid only at the sides of the thin piezo membranes, and the acoustic damping is minimum. In contrast to the other modes, the resonant frequency is not decided by the thickness of the piezoelectric film, but by the lateral dimensions of the device, which can be lithographically controlled [48,177].

##### 2.1.2.2. FBAR as Biosensors: Operating Principle

The basic operating principle of FBAR in biosensing is similar to that of QCM, as explained in Section 2.1.1.1. The sensor operates as a gravimetric sensor, which produces a shift in the resonant frequency proportional to the adsorbed mass on the surface, as represented by Equations (5) and (7). When operating in a liquid medium for biosensing, the basic gravimetric principle may not be sufficient to explain the measured frequency shift, as the sensor responds to the interfacial phenomenon as well as the viscoelastic properties of the medium [77]. A careful interpretation of the measurement results will help to bifurcate the contributions from the mass loading and the viscoelastic loading, due to the surrounding medium or the adsorbed complex biomolecules, such as the DNA oligonucleotides.

The major difference occurs in the area of functionalizing the FBAR with bioreceptors, for the selective detection of the target analyte from the sample. The bioreceptors used are similar to any of the acoustic biosensors, such as an antibody, aptamers, and nucleic acid probes [178,179,180]. The backside air cavity design enables the bioreceptors to be functionalized on the backside cavity, after coating the cavity area beneath the membrane with an Au sensing layer [181]. Another method of functionalization in the SMR and back cavity design, is utilizing the top Au electrode of the FBAR to immobilize the bioreceptors. Various bioreceptor immobilization schemes are shown in Figure 19a–c. the performance of the FBAR biosensors can be quantified using mass sensitivity and LOD, as expressed in Equations (6) and (10), respectively. The high resonant frequency of the FBAR devices, enables a 1000 times higher mass sensitivity and lower LOD in the range of picograms to femtograms [77,169,182].

##### 2.1.2.3. Challenges in FBAR Biosensing

One of the major challenges in using FBAR for biosensing, is the acoustic damping and reduction in the Q factor when operating in the TE mode. The reduction in the Q factor, affects the minimum detectable change in mass due to the acoustic energy loss. For a 1.5 GHz FBAR operating in TE mode, a Q factor as low as 15 was reported in liquid media, when compared to 250 when operating in the air [170]. The excitation of shear mode in FBAR with a good Q factor, is the requirement for FBAR biosensors operating in the liquid environment [46]. Operating FBAR in the TSM mode, by utilizing the c-axis tilted piezoelectric thin films and lateral field excitation (LFE), helps to minimize the acoustic dampening due to the shear mode acoustic waves [49,59].

Operating the FBAR at second harmonics has been reported to improve the Q factor in the liquid medium [185]. Another method to improve the Q factor when operating in a liquid environment, is to reduce the height of the liquid medium in contact with the sensor to less than one acoustic wavelength, by integrating a microfluidic channel on top of the FBAR [186]. Backside sensing with an integrated microfluidic channel at the backside, is also performed to improve the Q factor performance, as shown in Figure 18a [59]. Even though FBAR exhibits the highest resonant frequency of all types of acoustic biosensors with miniature dimensions, the operation in a liquid medium with the top side as the active area can pose difficulty in the bond pad connections [46]. The top electrode track to the pads must be made long enough to keep the bond pads away from the analytes, but keeping the parasitic resistance to a minimum [173]. The device yield rate, in terms of fabrication, is also a challenging factor due to the free-standing membrane structures. A careful balance between the stress of supporting the SiO_2_ or Si_3_N_4_ layer and the piezo thin films, must be achieved for the formation of crack-free membranes. Mass amplification methods, involving sandwich immunoassays with nanoparticles, are also incorporated for improving the mass sensitivity of FBAR biosensors to detect small molecular weight targets at lower concentrations.

##### 2.1.2.4. Piezoelectric Materials and Fabrication

Piezoelectric thin films, such as AlN and ZnO, are mainly used for FBAR biosensors due to biocompatibility and desired piezoelectric properties [1]. The film deposited in the FBAR backside air cavity design needs to have careful control of stress to form a rupture-free, free-standing membrane. Film properties, such as crystallographic orientation, thickness, and surface roughness, play a significant role in the Q factor and resonant frequency of the device.

Apart from the above-mentioned properties of the piezo thin films, the electromechanical coupling coefficient (K^2^), acoustic wave velocity in the thin films, and temperature coefficient of frequency (TCF) of the film, affect the biosensor performance. As FBAR devices are excited by applying high-frequency AC voltage, the electromechanical coupling coefficient plays a significant role in determining the energy transduction efficiency. Apart from the piezoelectric properties of the thin film, K^2^ is also affected by other factors like the electrode material and losses in the thin film [100]. The PZT film offers higher K^2^ due to the higher piezoelectric coefficient of the film, but leads to an increased acoustic energy loss, lower acoustic wave velocity, and lesser compatibility for biological applications [30]. Acoustic wave velocity is highest for AlN, resulting in better mass sensitivity. AlN film is CMOS compatible and can be integrated with CMOS electronics. They are generally deposited by physical vapor deposition (PVD). ZnO has a lower acoustic wave velocity compared to AlN and is not CMOS compatible, but has better K^2^ than AlN. ZnO film can be deposited either by sputtering or the Sol-gel method, and has better surface roughness compared to AlN.

TCF is also another important parameter that determines the stability of the biosensor response to temperature variations. As mentioned in Section 2, the acoustic wave velocity in the piezoelectric materials depends on the dielectric, piezoelectric, and elastic properties. Hence, the temperature variation of acoustic frequency depends on the temperature dependence of these properties [43]. TCF can be represented as [1]
(12)$$\mathrm{TCF}=\frac{1}{f_{0}}\frac{{\mathrm{df}}_{0}}{\mathrm{dT}}=\frac{1}{v}\frac{\mathrm{dv}}{\mathrm{dT}}-\alpha ,$$
where *f*_0_ is the resonant frequency, *T* is the temperature, *v* is the acoustic velocity, and α is the temperature coefficient of expansion. Generally, piezoelectric thin films have higher TCF, which requires a temperature compensation layer for the working of FBAR biosensors. The SiO_2_ compensating layer has been used as a supporting layer that has opposite TCF, compared to piezoelectric thin films [187]. Another method reported to reduce TCF, is the integration of an air gap metallic capacitor with FBAR to make the TCF reach 0 [188]. FBAR supporting dual modes with opposite responses to temperature variations, have also been reported for the improvement of TCF [189]. The major parameters related to the piezoelectric thin films are listed in Table 1.

The fabrication of FBAR device is conducted by using Si micromachining, to form the free-standing back cavity design. The typical top electrode metal used is Au, if the top side functionalization of the bioreceptor is performed. Other materials used for the electrodes are Al, Ti/Pt, and Mo [176,190,191]. Electrode material plays an important role in the FBAR performance. Materials with low mass density, high conductivity, and higher acoustic impedance mismatch to the piezoelectric material are needed to improve the Q factor [100]. Carbon nanotubes (CNT), as the top electrodes, have been reported to improve the Q factor and biosensor performance by increasing the surface area for bioreceptor binding [100,192]. The initial step in the fabrication of the FBAR device, is the deposition of supporting membrane, either Si_3_N_4_ or SiO_2_ followed by bottom electrode deposition and patterning. Additionally, the piezoelectric thin film, either AlN or ZnO, is deposited. The deposited piezoelectric layer is then patterned, and vias are formed for making contact with the bottom electrode. The top electrode layer is then deposited and patterned, which forms the front side of the FBAR. The backside air cavity is etched by the wet etching of Si using KOH or by dry etching using DRIE [193]. The typical fabrication process flow and completed device for a backside cavity FBAR device operating in the TE mode, are shown in Figure 20a,c, respectively. An extra layer of Cr/Au or Ti/Au electrode is deposited on the backside cavity membrane area for biosensing, as shown in Figure 20b, respectively. The device can be integrated with microfluidic channels for sensing in a liquid medium [173]. Similar fabrication steps are applicable for SMR FBAR devices, except for the need for the continuous deposition of the thin film for Bragg reflectors and the absence of backside cavity etching.

##### 2.1.2.5. Applications of FBAR in Biosensing

The higher resonant frequency of the FBARs, compared to QCM, makes them attractive for biosensor applications due to the improvement in sensitivity. The bioreceptors for the detection of target analytes can be antibodies, aptamers, or oligonucleotide probes, similar to QCM biosensors. FBAR biosensors have been utilized for detecting various complex proteins, biomolecules, and nucleic acid molecules, and for studying the complex behaviors of protein interactions on the sensor surface. The following section describes some of the applications of biosensors in detecting the above-mentioned target analytes. Appendix A, lists some of the applications of FBAR for sensing complex biomolecules with their performance index, reported in the past decade.

(i)Proteins and Biomolecular Detection

FBAR biosensors have been utilized for the detection of complex small molecular weight biomolecules, due to their extremely high mass sensitivity [169]. They are used for the detection of various disease biomarkers in human serum, such as the carcinoembryonic antigen (CEA), prostate-specific antigen (PSA), alpha-fetoprotein (AFP), epithelial tumor marker mucin 1(MUC-1), and allergic antigen, immunoglobulin E(IgE) [179,180,190,194,195,196,197]. The bioreceptors commonly used are either antibodies or aptamers specific to the target antigen.

The rapid and label-free detection of CEA with high sensitivity, has been reported using the FBAR biosensor with both backside air cavity design and SMR design [173,180,183,198]. The SMR with c-axis 20° tilted AlN film operating at 2 GHz in TSM, was demonstrated for the detection of CEA using aptamers as bioreceptors [173,180]. The detection of CEA by the aptamer resulted in the lowering of the resonant frequency in SMR, due to mass addition. The device was reported to have a mass change sensitivity of 2045.89 Hz cm^2^/ng.

Mucin 1 (MUC1) is a protein produced by human epithelial tissues, whose overexpression in blood serum can be a biomarker for cancer affecting various parts, such as the lungs, breast, pancreas, or bladder [199]. FBAR in SMR configuration, with the AlN and ZnO piezoelectric layer, has been reported for the detection of MUC1 [195,200]. SMR with a resonant frequency of 1.5 GHz and a c-axis oriented ZnO film on top of Ti/Mo alternate layers of λ/4 thickness as the Bragg reflector exhibited a mass sensitivity of 4642.6 Hz/nM and an LOD of 20 nM for MUC1 detection [200]. An indirect detection method of target MUC1 was demonstrated based on the biotin–avidin binding on the SMR surface. The MUC-1 captured AuNP-aptamer conjugate with exposed biotin, when introduced to the streptavidin immobilized FBAR sensor, resulted in a frequency drop due to the mass addition via biotin–avidin binding. In the absence of MUC-1, the aptamer remained closed without exposing the biotin and, hence, did not bind to FBAR sensor, as shown in Figure 21a. The frequency response of the FBAR sensor with each step of surface mass addition, is shown in Figure 21b.

FBAR biosensors are also used for the study of protein–ligand interactions and the detection of proteins and biomolecules, such as thrombin, glucose, and pesticides, making them a promising candidate for LOC and POC devices, as well as for industrial applications [15,46,176,193]. The detection of thrombin was demonstrated using SMR FBAR with a 25° tilted c-axis AlN with SiO_2_/TaO_x_ fully insulating alternating layers of non-λ/4 thickness as Bragg reflectors, using the aptamer as a bioreceptor [46]. The use of SiO_2_/TaO_x_ alternating layers was performed, to minimize the parasitic capacitance effect introduced while extending the contact pads away from the sensing area. The modification of the alternating layer Bragg reflector thickness was conducted to improve the TCF. The device schematic is shown in Figure 22a. Shear mode excited the device with iridium (Ir), as the top and bottom electrodes exhibited a mass change sensitivity of 1800 kHz/pg·cm^2^ at 1.3 GHz resonant frequency. The device showed a decrease in frequency, with an increase in the concentration of target thrombin due to mass addition, as shown in Figure 22b.

Apart from detecting targets by the mass sensing principle, FBAR devices have also been reported for monitoring the viscosity of the target species, such as blood coagulation monitoring in real-time by frequency monitoring [49,201]. Blood and plasma coagulation kinetics, in the presence of the coagulating agent, thromboplastin (TP), was monitored in real-time by using SMR FBAR with lateral field excitation [49]. Coplanar Au electrodes on the c-axis oriented AlN was used for exciting pure shear mode at 1.9 GHz, which enabled device operation in the liquid environment. The polymerization of fibrin, due to blood coagulation, increased blood viscosity, which was determined by a gradual drop in the resonant frequency of the FBAR sensor. The FBAR resonator showed a promising performance for monitoring blood coagulation with a very small quantity of blood sample within the small dimension of 500 µm × 500 µm of the sensor area [49].

(ii)DNA Biosensor

The high mass sensitivity of FBARs makes them suitable for sensing low concentrations of DNA, by hybridization with an immobilized probe oligonucleotide on the sensor active area as the bioreceptor. This makes FBAR a good choice for DNA biosensors. Sequence-specific, label-free detection of DNA hybridization in complex mediums, such as human serum and the real-time monitoring of DNA synthesis, has been demonstrated using FBAR biosensors [178,181,182,202].

The SMR FBAR biosensor with a ZnO thin film sandwiched between the Au electrodes operating at 800 MHz, has been reported for the detection of DNA hybridization of short oligonucleotide strands at nM concentrations from the buffer medium [178]. To minimize the non-specific binding (NSB) of non-complementary strands onto the sensor, and to improve the device specificity and sensitivity, the Au top electrode was coated with lipoamide (Lipa-DEA), after functionalization with the capture probe DNA bioreceptor layer. The hydrophilic coating of Lipa-DEA also helped in the proper orientation of probe DNA, which improved the hybridization efficiency of the DNA sensor. The FBAR sensor was reported to successfully detect long DNA strands from 1% diluted serum at nM concentrations. DNA synthesis, using polymerase chain reaction (PCR), has also been monitored at real-time using a ZnO FBAR resonator with an immobilized probe ssDNA and a target template DNA sequence [202].

### 2.2. Surface Acoustic Wave Biosensors

Surface acoustic wave derives its name as a result of the energy being concentrated on the surface of the piezoelectric substrate, within one to two acoustic wavelength thicknesses, with the acoustic amplitude decaying along the thickness direction [43,203]. Wave propagation through the surface of the elastic solids was first proposed by Lord Rayleigh, in 1885 [204]. Since then, surface waves have been extensively studied in relation to seismic waves, in which mechanical waves propagate along the surface of the earth [205]. The first use of periodic electrode arrays to directly excite elastic surface waves on a piezoelectric substrate, was demonstrated by R. M. White and F. W. Voltmer [206]. This discovery led to the application of surface acoustic wave (SAW) devices in signal processing [207].

For the generation of SAWs, interdigitated transducers (IDT), as shown in Figure 23, are used. These periodic electrode arrays are deposited on piezoelectric substrates or thin films, where the dimensions and number of fingers determine the frequency, insertion loss, and bandwidth of the device [208]. The electrical activation of the IDTs induces compressions and rarefactions underneath the IDTs, according to the charge introduced, resulting in a mechanical acoustic wave and electric field across the surface [43]. For the applied RF frequency, *f*, the generated mechanical waves constructively interfere if the pitch of the IDT represented by *p*, in Figure 23, is equal to the acoustic wavelength. The corresponding frequency of the propagating surface wave is the resonant frequency of the device, given by [203,209]
(13)$$f=\frac{v}{p},$$
where *v* is the velocity of the SAW on the chosen piezoelectric substrate and *p* is the pitch of IDT. SAW devices can be designed to operate between 10 MHz to 2.5 GHz on piezoelectric substrates, such as quartz, LiTaO_3_, and LiNbO_3,_ and thin films, such as AlN, PZT, and ZnO [56,210,211,212,213,214,215,216]. This operating frequency is almost more than 100 times higher than that of bulk waves in QCM.

SAW devices have significant advantages in electronic devices, in terms of miniature size, low cost, ease of manufacture, and all electrical readout, which makes them suitable for applications, such as resonators [218]; filters [219,220]; radio frequency identification [221]; chemical sensors [222,223,224,225,226]; pressure sensors [227,228]; temperature sensors [229,230,231]; strain sensors [232]; and biosensors [40,233,234]. The first reported SAW-based sensor was in 1979, which was used for chemical gas sensing using both quartz and LiNbO3 substrates [235]. The high surface energy density of the SAW makes them highly sensitive to any surface perturbations, making them an excellent candidate for sensors. In this review, we discuss the most recent applications for SAW devices as biosensors and microfluidic devices.

#### 2.2.1. SAW Modes for Biosensors

Since acoustic waves associated with SAW devices are mechanical elastic waves propagating on the surface of piezoelectric substrates, various modes can be defined based on the particle displacement trajectory. The SAW propagation modes can be broadly classified as surface-based and plate-based waves, depending on the confinement of the acoustic energy on the piezoelectric substrate. Depending on the direction of particle displacements on the surface, with respect to the wave propagation direction and the substrate surface, surface-based SAW can be further classified into the Rayleigh wave and Love wave modes. Plate-based waves travel within the thin plates or membranes with a thickness varying from a few percentages to a few acoustic wavelengths, and are generated using IDTs [236]. They can be further classified into the Lamb wave and shear horizontal acoustic plate mode (SH-A PM). These different mode types are determined by the piezoelectric substrate material, substrate thickness, crystalline orientation of the substrate, and orientation of IDT, with respect to the crystallographic axis of the substrate [236]. Figure 24 shows the schematic representation of the SAW propagation modes.

As biosensing is often performed in a liquid medium, not all these modes are suitable as the particle displacement has components normal to the substrate surface. SAW propagation in the liquid medium, also results in a loss of acoustic energy and, hence, attenuation occurs [1,236]. The following section describes various SAW modes and their usefulness in biosensing. A compilation of the SAW device structure, wave propagation modes, substrates used, operating frequency, and substrate thickness for the modes, are shown in Table 2.

(i)Rayleigh Mode SAW (R-SAW)

The particle displacements, here, have both surface parallel components and surface normal components. This results in an elliptical trajectory for the surface particles, as seen in Figure 24. This mode is not suited for sensing in the liquid medium, due to the acoustic attenuation caused by the coupling of compression waves to the medium by the surface normal component [236]. This mode is more often used for microfluidic acoustic actuation devices, which induces an acoustic streaming force when acoustic attenuation occurs due to coupling with the contact fluid in the delay line path. However, some biosensors have been proposed that operate on the “dip and dry” method and resonator configurations [237,238].

(ii)Love Wave Mode SAW (Shear Horizontal (SH) SAW Love)

The particle displacements for the shear horizontal mode (SH-SAW) are parallel to the piezoelectric substrate surface and perpendicular to the direction of wave propagation, as given in Figure 24. These waves are more pronounced than Rayleigh waves for thick piezoelectric substrates with different crystal orientations, as shown in Table 2. The bulk waves excited in the piezoelectric substrate are made to concentrate on the substrate surface for Love mode, by coating a guiding material on the substrate surface, providing the highest sensitivity out of all the modes. This makes the Love mode a guided SH-SAW. The IDTs used in this case are sandwiched between the interface of the guiding material and the substrate. The guiding material has to have a lower acoustic wave velocity than the piezoelectric substrate, causing the wave to be concentrated and confined to the guiding layer, making it more vulnerable to surface perturbations. The thickness of the guide layer is also designed to enhance sensitivity and minimize acoustic attenuation [239,240]. Biocompatibility and corrosion resistance are also considered. The commonly used biocompatible guiding materials are SiO_2_ [241], ZnO [242], Parylene-C [50], polymers like polymethyl methacrylate (PMMA) [243], polyimide [244], and photoresists, such as Novolac [240]. A detailed review of the guiding materials used for the Love wave mode biosensors is covered by other researchers [245].

(iii)Lamb Wave Mode SAW

Lamb waves are equivalent to Rayleigh waves in their thin plate-like substrates instead of the bulk substrates, in which the plate thickness is slightly higher than two acoustic wavelengths. The wave extends into the bulk of the thin plate, similar to the BAW, but is generated by using IDTs. Two Rayleigh waves exist in the plate, guided between the top and bottom surface of the thin plate and result in forming symmetric and antisymmetric modes in the plates. Of these modes, the lowest order antisymmetric mode (A_0_ mode) has a wave velocity that is much lower than the corresponding velocity of the SAW in the same medium. The wave velocity of the A_0_ mode decreases as the plate thickness decreases, and becomes more concentrated on the plate surface. These modes are also termed as flexural plate wave modes (FPW) [236]. FPW devices are generally realized by using very thin film piezoelectric films, such as ZnO, AlN, and PZT, forming membrane-like devices with the thickness of the plate being only a small percentage of the acoustic wavelength [212,216,246,247]. Even though the particle displacements for the FPW mode have surface normal components, they are suitable for sensing in the liquid medium, given that their wave velocity is much lower than the compressional velocity in the liquid medium. Another advantage of using an FPW device, is that, since the acoustic wave extends into the bulk of the thin membrane, both sides of the membrane are suitable for sensing [212]. The possibility of the contamination of and corrosion to the IDT can also be avoided and electronics can be separated from the sensing medium.

(iv)Shear Horizontal Acoustic Plate Mode (SH-A PM)

SH-A PM waves are shear horizontal acoustic waves in the thin plate-like substrates, with particle displacements parallel to the substrate surface. Similar to Lamb wave devices, the SH-SAW waves in the A PM devices extend to the bulk of the thin plates, so that both the top and bottom surfaces are suitable for sensing [248]. Different modes coexist within these SH-A PM devices, and the careful design of the transducer enables the device to operate in a single mode [249]. The device sensitivity for biosensing increases as the plate thickness reduces, due to an increase in the mode frequency and, hence, a higher wave velocity resulting in a greater concentration of acoustic energy at the surface [239].

#### 2.2.2. SAW as Biosensors: Operating Principle

Biosensors using SAW are generally found using two configurations. One is the delay line configuration, as shown in Figure 25a. Two IDTs are fabricated at the opposite end of the piezoelectric material, at a specific distance from each other. The waves travel with their phase velocity on the piezoelectric surface between the two IDTs and reach the receiver IDT after a time delay, hence its name. The electric field associated with the acoustic waves, generates a voltage at the receiving IDT [43,239]. This enables the characterization of the SAW properties, such as wave velocity and attenuation, by measuring the amplitude, phase, and frequency of the SAW at the receiver IDT.

Another possible configuration for the SAW sensor is the resonator, as shown in Figure 25b. A set of reflectors on either side of the IDT, enables the surface waves to be reflected and contained in the resonator. Resonator mode can help to increase the sensitivity of the biosensor, due to an increase in the wave propagation path [250]. Both these configurations are either deployed in liquid sensing or incorporated with microfluidic channels.

Biosensors based on SAW devices, have the advantage of being label-free, small in size, low cost, easy to fabricate, and able to operate as a wireless device. Since biosensing is performed in liquid media, the ability to incorporate SAW-based devices with microfluidic channels, makes them a good choice for sensing biological entities in point of care applications [251]. The basic principle of operation involves functionalizing the delay line between IDTs, by coating with a sensitive layer, such as gold, graphene oxide, or single-layered graphene (SLG), which act as binding sites for the bioreceptors [55,214,252].

In Love mode SAW biosensors, this sensing layer is either deposited on the waveguide material between the IDTs, or the waveguide layer itself becomes the sensing surface [253,254]. For FPW mode, the coating of the sensitive layer is made at the backside of the membrane in the cavity [212]. The standard architecture of a crystalline substrate-based SAW biosensor in delay line configuration in SH-SAW mode, FPW mode, Love wave mode, and resonator configuration, is shown in Figure 26a–d. SAW delay line biosensors utilizing a piezoelectric thin film, is shown in Figure 26e. The sensitive layer enables the immobilization of bioreceptors. The bioreceptors can be antigen, antibody, nucleic acid probes, synthetic aptamers, and molecularly imprinted polymers, which are bound to the sensing layer, either by covalent bonding or by physical adsorption mechanisms, such as electrostatic or hydrophobic interactions [45,255,256]. The bioreceptors will selectively and specifically adsorb the target analyte from the sample, enabling label-free sensing and repelling other background molecules that are present.

Sensing is achieved by the perturbation of the SAW in the delay line, due to mass loading by the bound target analyte, viscoelastic loading of the sample solution, or by the electrical conductivity of the sample in contact with the sensitive layer. This perturbation affects the wave velocity and damping of the traveling acoustic waves. This can be detected at the receiver IDT as phase delay, amplitude attenuation, or as the frequency shift of the acoustic waves, which corresponds to the electrical equivalent of the detected target properties, as described in Section 1. The change in the acoustic wave velocity (*v*) is related to the frequency (*f*) and phase shift (*Φ*) of the received signal, as given by [257]
(14)$$\frac{∆v}{v}=\frac{∆f}{f}=-\frac{∆\emptyset }{\emptyset },$$

The typical operation of the SAW biosensor as a gravimetric sensor, results in the lowering of the resonant frequency, which is proportional to the added mass. The frequency response of the delay line device can be measured by using a network analyzer, which provides the impedance characteristics as a function of the frequency [239]. A vector voltmeter can also be used for measuring the amplitude and phase difference of the delay line signals. More precisely, highly sensitive measurements can be made by measuring the resonant frequency shift by operating the SAW device in the feedback loop of an RF amplifier forming an oscillator circuit [203].

The performance of SAW biosensors is expressed in terms of mass sensitivity and limit of detection (LOD) [80,258]. The mass sensitivity, *S_m_*, of a SAW biosensor is defined as the output frequency change *(*∆f), due to the input adsorbed mass per unit area (∆m), and is proportional to the square of the SAW resonant frequency [259]. Sensitivity is experimentally calculated from the slope of the calibration curve of the biosensor, and is expressed in Equation (6). The LOD represents the smallest concentration of the target analyte that can be detected by the sensor, and can be calculated similarly as used in QCM, as mentioned in Equation (10), in Section 2.1.1.1. A lower LOD is required for biosensors. Since SAW devices have operating frequencies higher than bulk waves, the sensitivity is much higher than QCM and a lower limit of detection, down to µg/mL to tens of ng/mL, is possible [237]. An improvement in the SAW biosensor sensitivity and the lowering of LOD, has been demonstrated by utilizing mass amplification methods, involving sandwich immunoassay, similar to BAW biosensors [258,260].

#### 2.2.3. Piezoelectric Materials and Fabrication

Ever since the discovery of direct surface wave generation in piezoelectric substrates using IDT, the various modes in surface acoustic waves are generated by carefully choosing various piezoelectric substrates. In recent years, substrates like quartz, lithium niobate (LiNbO_3_), and lithium tantalate (LiTaO_3_), are found to be more commonly used for SAW biosensors [264,265,266]. The SAW modes and wave velocity is determined by the substrate parameters, such as the crystal cut, orientation, and substrate thickness, and the frequency is determined by the IDT dimension [236]. The piezoelectric substrate parameters that influence the SAW properties are the electromechanical coupling coefficient (K^2^), temperature coefficient of frequency (TCF), and dielectric constant [264]. Since piezoelectricity is exhibited by anisotropic solids, these parameters are also crystallographic orientation dependent.

The electromechanical coupling coefficient, determines the efficiency of energy conversion between the electrical signal and mechanical wave in the piezoelectric substrate. As an example, LiNbO_3_ has a higher K^2^ value compared to that of quartz, as shown in Table 1, which implies that to excite the SAW in quartz, a greater number of IDT finger pairs is required compared to that in LiNbO_3_. The temperature coefficient of frequency (TCF), determines the variation of the SAW wave velocity and, hence, the synchronous frequency of the device with respect to the temperature variations [265]. This is also due to substrate material and crystallographic properties. TCF variation can also be overcome, by incorporating heat absorbers into the SAW device [267] or using a dual-channel delay line as a control reference [253]. The dielectric constant difference between the substrate and medium, can lead to reduced electroacoustic coupling and the attenuation of the SAW energy [268]. LiNbO_3_ and LiTaO_3_ have higher dielectric constants compared to quartz, enabling better compatibility for operating in a liquid medium. The choice of piezoelectric material for SAW biosensors, requires the consideration of all the above-mentioned parameters that can be summarized as higher electromechanical coupling coefficients, lower TCF, and the reduced dielectric constant mismatch between substrate and liquid samples.

Apart from the bulk substrates, deposited piezoelectric thin film materials, such as ZnO, AlN, PZT, and PVDF, on Si, diamond, sapphire, SiC, and polymer substrates, have also been used for the SAW generation for biosensors [62,269,270]. The higher electromechanical coupling coefficient and higher acoustic wave velocity of the thin films, make them an excellent candidate for SAW biosensors by improving the sensitivity [1]. Thin films also enable the fabrication of flexible SAW-based biosensors [56]. One advantage of thin-film SAW biosensors is that CMOS-compatible devices can be formed with integrated electronics. A more in-depth review of thin films used for biosensing can be found here [1]. The values of K^2^, TCF, and the wave velocity of commonly used substrates and thin films for biosensors, are shown in Table 1.

The fabrication of SAW biosensors involves the evaporation of metal electrodes on the piezoelectric substrate or thin film, followed by patterning the IDT and functional layer (delay line) using planar IC technology. The typical electrode material used is a combination of Cr/Au or Ti/Au or Al. If Al is used, then a protective coating of polymer material or SiO_2_ is needed to prevent the corrosion of Al from the biosensing medium [215]. Au is the most commonly preferred electrode material due to its biocompatibility and chemical stability. Cr or Ti is used as an adhesion promoter. The dimension of the IDT is determined by the resonant frequency of the SAW device, as well as the impedance matching requirement of the sensor surface. Fabrication of IDT is performed either by conventional UV photolithography or by electron beam lithography, depending on the dimensions.

A typical process flow for the fabrication of a Love wave biosensor, is shown in Figure 27a. An additional process step for the deposition of the guide layer is required for the Love mode devices. Plasma enhanced chemical vapor deposition is mostly used for the deposition of the SiO_2_ guiding layer, PVD for ZnO, and spin coating for polymer guide layers. A fabricated Love wave device with an Au electrode, sensing active area, and SiO_2_ guiding layer, is shown in Figure 27b. the deposition of the guide layer is followed by patterning and etching, to selectively remove the guide layer to open up the bond pads for connections to the IDT. For the biosensing applications for LOC and POC, SAW devices are integrated with either PMMA flow cells or PDMS microfluidic channels for sensing in the liquid medium [262,271]. An integrated SAW device with a microfluidic channel for biosensing is shown in Figure 27c.

The fabrication of a Lamb wave (FPW) device is complicated, as it involves the use of silicon micromachining to form thin plates and membranes. Silicon dioxide (SiO_2_) or silicon nitride (Si_3_N_4_), is used as an epitaxial supporting membrane for the sputtering of ZnO/AlN/PZT as a piezoelectric layer. Apart from being a supporting membrane that acts as an etch stop layer for Si back cavity etching, these layers also provide electrical isolation. The fabrication steps of a Lamb wave device with a ZnO piezoelectric thin membrane, are shown in Figure 27d. The floating membrane of the device is made of the stack Si/SiO_2_/Si_3_N_4_/Cr/Au/ZnO [247].

#### 2.2.4. Applications of SAW in Biosensing

SAW velocity is perturbed by the mass attached to the surface along the direction of the acoustic wave. Simultaneously, the changes in the viscoelastic properties and conductivity of the adsorbed mass and the medium in contact with the surface, make it an interesting candidate for label-free biosensing. Additionally, the low attenuation of horizontally polarized waves in a liquid medium without considerable acoustic attenuation also provides an added advantage [268,275]. The application of SAW devices in biosensing can thus be extended to multiple fields, such as biomedical applications for detecting a variety of disease biomarkers, tumor cell detection, the study of cell adhesion, detection of pathogens and viruses, and complex biomolecules, including lipopolysaccharides for drug quality test, DNA, levels of glucose, and uric acid in human serum [18,52,55,211,214,276,277,278]. The bioreceptors can be an antibody, aptamers, or MIPs, or can be oligonucleotides, such as peptide nucleic acid or single-stranded DNA (ssDNA) [254,276,279,280]. The following section focuses on the application of SAW biosensors for the detection of various analytes. A brief overview of the SAW modes with their corresponding applications, design details surface functionalization for bioreceptor, and detection mechanism, is listed in Appendix A.

(i)Protein and Biomolecular Detection

The detection of proteins and biomolecules in human serum plays an important role in the clinical diagnosis of diseases. Biomarkers, which are proteins found in human serum, for the early detection of inflammations, allergic reactions, diseases like cancer, neurological disorders, myocardial infarction, and viral infectious diseases can be detected using SAW biosensors [247,258,278,281,282,283]. Sensitive, rapid, real-time, label-free, and easy detection of these disease biomarkers without the use of extra reagents and complex laboratory expertise or equipment, makes them suitable for POC applications.

Viral infectious disease biomarkers in human serum can be used for the early diagnosis of the human immunodeficiency virus (HIV), avian influenza (AI), and respiratory syncytial virus (RSV). SAW biosensors have been reported for the detection of viral disease biomarkers for HIV, such as anti-p24 antibodies (against viral capsid p24 protein), anti-gp41 antibodies (against gp41 glycoprotein), and for AI H5N1, such as the hemagglutinin (HA-surface protein in H5N1 virus) antigen [54,283,284]. They have been demonstrated for POC applications with real-time and rapid detection capabilities. An FPW device using PZT has also been demonstrated for the detection of RSV, via monitoring the capture of signaling protein chemokines in serum with a sensitivity of 14 Hz/nM [216,285]. The FPW membrane thickness here was 3.8 µm, with the backside of the FPW cavity being used as the active sensing layer. The capture of target chemokines resulted in a frequency reduction of the FPW sensor due to mass loading.

The SAW biosensor with either a ST-quartz or LiTaO_3_ substrate, has been reported for the detection of biomarkers for the early diagnosis of cancer, such as the carcinoembryonic antigen (CEA), prostate-specific membrane antigen (PSMA), epidermal growth factor (EGF), B-cell lymphoma, and exosomes [213,252,253,258,260,271,273,286].

Love mode and FPW SAW mode sensors have been reported for the real-time, label-free detection of CEA, from both human serum as well as from exhaled breath condensate with an LOD varying from 1 ng/mL to as low as 37 pg/mL [258,261,271,273,287]. A Love mode SAW delay line biosensor using ST-cut quartz with Au IDT and sensing area operating at 120 MHz was reported, with an LOD of 0.31 ng/mL for CEA detection [287]. The same group subsequently reported a superior method for the formation of an anti-CEA antibody bioreceptor layer on the same SAW sensor, for the detection of CEA with an improved LOD of 0.084 ng/mL [252]. The anti-CEA antibody layer was immobilized on a conducting polymer coating with a graphene oxide (GO), MoS_2,_ Au nanoparticle (AuNP) cluster on the Au sensing area. The GO-MoS_2_ nanomaterials helped to reduce the insertion loss in the polymer coating and the AuNP greatly improved the anti-CEA antibody adsorption density. A different signal amplification strategy using a sandwich immunoassay is also reported here, in which a much lower detection limit of 37 pg/mL for CEA was observed using a Love mode SAW sensor with ST quartz and 1 µm SiO_2_ waveguide layer with a resonant frequency of 120 MHz [258]. The device schematic is shown in Figure 28a,b, with an integrated PDMS microchannel. The real-time monitoring of phase shift showed an enhanced phase change after the addition of nanoprobes due to mass amplification, as reported in Figure 28c.

An FPW sensor operating at 20 MHz was reported for the detection of CEA utilizing 1 µm c-axis oriented ZnO thin film with a detection limit of 5 ng/mL [261]. A novel fan-shaped and circular IDT and reflectors were introduced to minimize the insertion loss (IL) in the device, due to the loss of acoustic energy to the bulk Si substrate in the FPW device. The device schematic and cross-section for both IDT geometry are shown in Figure 29a. The anti-CEA antibody was chemically bound to the sensing Au area at the cavity backside, which selectively and specifically captured the target. The frequency shift increased with an increase in the concentration of CEA, as shown in Figure 29b.

SH-SAW sensors have also been reported for the detection of cardiac markers, such as cardiac troponin I (cTnI), creatine kinase (CK)-MB, and myoglobin [281,288]. The detection of cardiac markers in human plasma is considered as “gold standard” for acute myocardial infarction (AMI) [289]. The Love mode biosensor was reported to detect cTnI with a detection limit of 6.2 pg/mL [281]. The device used 36° YX-cut LiTaO_3_ with Al IDT and a 5.2 µm thick SiO_2_ as a guiding layer, operating at a resonance frequency of 200 MHz. The device was designed with a working channel and a reference channel to cancel out the interference effects, as shown in Figure 30a. Sandwich immunoassay with an AuNP conjugated detection antibody of cTnI along with Au staining was used as a mass enhancement technique to improve sensitivity and to lower the LOD, as schematically shown in Figure 30b. A normalization technique was employed for estimating the sensor response, with the ratio of performance of the working sensor to the reference sensor as a performance index to improve the reproducibility of the immunosensor and to cancel out the interference.

Apart from being used as immunosensors for detecting antibody–antigen interactions, SAW biosensors have also been used for the detection of proteins and biomolecules in the human body. Neurological and mental disorders, such as Parkinson’s, Alzheimer’s, or schizophrenia, can be detected by monitoring amino acids in human serum, which act as neurotransmitters [290,291]. Dopamine, d-serine is in such proteins whose detection has been demonstrated using the SAW resonator as well as with SAW delay line configurations [262,278,292,293]. Molecularly imprinted polymers (MIP) or enzymes were used as bioreceptors. Label-free detection of dopamine (DA) has been reported using a Love mode sensor on ST cut quartz with an LOD of 0.1 pg/mL [293]. The electrochemical deposition of MIP with a cerebral dopamine neurotrophic factor (CDNF) as the template, was performed on the Au sensing area of the Love wave device utilizing a cleavable linker. A high sensor specificity ratio of 9:1 for the CDNF target as compared with the control was observed with the target-specific cavities present in MIP.

Apart from the disease biomarkers, SAW biosensors have also been reported for the detection of complex biomolecules, such as toxins, glucose, and uric acid in human serum; intracellular pH monitoring; and tetrameric enzymes used in cancer treatment [18,52,211,277]. Marine toxins, such as okadaic acid (OA) and bacterial toxins, such as endotoxin, have been detected using a Love wave sensor [55,294]. The SH-SAW aptasensor realized on a 36° Y–90° X quartz substrate with Cr/Au IDT operating at 246.2 MHz, was reported for *E. coli* endotoxin detection, with an LOD of 3.53 ng/mL [55]. The sensing layer utilized was a single-layer graphene oxide (SLG) instead of Au, which increased the device sensitivity and formed a biocompatible material for the immobilization of bioreceptors. A schematic of the functioning device with an immobilized aptamer detecting target endotoxin, is shown in Figure 31a. The real-time monitoring of the phase of the biosensor showed a shift in response to the endotoxin addition, due to the added mass, with an increase in phase shift with an increased concentration of target endotoxin, as shown in Figure 31b.

(ii)Cell Adhesion and Detection

SAW sensors are employed for the study of the quantification of cell growth in culture media, cell–cell interactions, tissue healing, and cell adhesion onto a substrate [295,296,297]. These play an important role in monitoring cellular activity and are useful in cell culture and tissue engineering. Cells adhered onto the sensors can also be used as bioreceptor layers forming cell-based sensors. Apart from the cell interaction studies, the detection of certain types of cells in human serum, such as circulating tumor cells (CTC), is useful in the clinical diagnosis of diseases [263,296].

The use of surface acoustic waves for dynamic stimulation and wound healing was demonstrated, which showed that cells exposed to SAW exhibited an increased cell growth and migration [295]. The Love wave sensor using 36° YX LiTaO_3_ substrate with Ti/Au/Ti IDT and SiO_2_ waveguide layer operating at 207 MHz, has been reported for the study of artificial wound healing [298]. The dynamic process of the motion of already adhered epithelial cells onto a substrate was demonstrated using a SAW sensor. The viscoelastic effect imposed by the cell coverage on the sensor resulted in the phase shift of the transmission coefficient (S_21_). The same device setup was also used to demonstrate the cell detachment assay involving cell apoptosis (cell death), with the phase shift showing the opposite trend when compared to the wound healing assay [298].

The study of cell-substrate adhesion plays an important role in cellular functions, such as cell migration, tissue engineering, and cell differentiation [299]. Monitoring cell-substrate adhesion can be performed using the SAW device by exploiting the sensitivity of surface waves to the viscoelastic property of adhered cells [297]. A Love wave device using 36° YX-LiTaO_3_ with Cr/Au IDT operating at a frequency of 131 MHz with parylene-C as the waveguide layer, was reported to be used for the real-time monitoring of the adhesion of tendon stem cells (TSC) to the substrate surface [300]. Cell adhesion on the sensor surface involved the collagen–TSC integrin protein interaction. The sensor parameters, such as insertion loss (IL) and phase, responded to the viscoelastic loading imposed by the adhesion of TSC from the suspension onto the sensor surface.

Apart from monitoring cell growth, cell adhesion, and cell migration on the sensor surface, SAW sensors are also used for the detection of circulating tumor cells (CTC), lymphocyte cells forming a cellular immune sensor [263,296,301]. Antibodies or synthetic aptamers specific to surface antigens present on the cells were used as bioreceptors. A leaky SAW device in the resonator configuration in a 2 × 3 array, was reported to detect the CTC with a detection limit of 32 cells/mL [263]. CTC in human serum plays an important role in determining the metastatic progression of cancer [302]. A 2-port resonator using 36° YX LiTaO_3_ with Cr/Au IDT, reflectors, and Au sensing area was designed to operate at 100 MHz with schematic, as is presented in Figure 32a. An aptamer specific to cancer cell surface protein mucin (MUC1), was functionalized on a working sensor with a random aptamer sequence in the reference sensor, as shown in Figure 32b. Real-time monitoring showed that the phase shift increased with an increase in the concentration of the MCF-7 cell, as shown in Figure 32c.

(iii)DNA Biosensors

SAW DNA biosensors have been used in the detection of gene sequences, single nucleotide polymorphism, and pathogenic microorganisms [47,303,304]. The detection is conducted by the hybridization of the probe DNA immobilized on the SAW biosensor, with the complementary target sequence that results in either mass loading or change in the viscoelastic properties of the adsorbed layer, due to the complex nature of the DNA molecules. SAW DNA biosensors provide real-time, label-free, sensitive, simple, and rapid detection compared to conventional schemes like PCR, ELISA, or electrochemical sensing methods, which require complex reagent steps and specialized handling expertise.

The detection of gene sequence plays an important role in deciding treatment protocols as well as in disease prevention, thereby contributing to the molecular diagnostics field [305]. SAW sensors with immobilized probe ssDNA as bioreceptors, have been demonstrated for DNA sequence detection by the mass loading effect [276,304]. The highly sensitive and specific detection of a DNA sequence with a very low detection limit of 0.8 pM has been reported using the commercially available SAW sensor sam5 system [276]. ST quartz with Au IDT and SiO_2_ waveguide layer, with Au coating for the sensing area, was used as a sensor. An enzyme mediated DNA strand elongation of the hybridized target, followed by an in situ Ag nanoparticle synthesis, was reported as a sensitivity enhancement method. A lower LOD, with 3 orders of magnitude lower, was realized compared to the direct detection method.

Single nucleotide polymorphism (SNP), where the gene sequence variation in DNA occurs when a single nucleotide is altered, plays an important role in determining the individual’s reaction to drugs and susceptibility to certain diseases [306]. SAW biosensors have been reported for the detection of base mismatches as well as SNP, due to the high sensitivity of the device [47,307,308]. Single nucleotide polymorphism (SNP) in a drug metabolism enzyme gene CYP2D6*10, in clinical samples, was detected by using a commercially available SAW biosensor sam5 [307]. The Au sensing area of the SH-SAW sensor was coated with graphene oxide (GO), which provided a biocompatible surface for the immobilization of capture probe ssDNA. The surface functionalization with probe DNA, and its interaction with various mutants of the target, is schematically represented in Figure 33a. The different mismatch sequences (fully complementary FC, single base mismatch SBM, and non-complementary NC) resulted in different changes in the phase shift, due to the variation in the mass and changes in the conformational properties, as shown in Figure 33b. The sensitivity, selectivity, and performance of the assay were demonstrated with direct sequencing and the device showed an LOD of 86.6 pM.

Apart from detecting gene sequences and gene mutations, SAW biosensors are also used for the detection of pathogenic microorganisms via the hybridization of the bacterial or viral gene sequence with immobilized complementary probe DNA as bioreceptors. SAW biosensors have been reported for the detection of gene sequences of pathogenic viruses, such as the human papillomavirus (HPV) and *Staphylococcus aureus*, and pathogenic bacteria, such as *Pseudomonas Aeruginosa*, *Salmonella*, and *E. coli* O157:H7 [214,303,309,310,311]. A Love wave sensor on ST-90° quartz, with Cr/Au electrode for IDT and SiO_2_ waveguide layer operating at 282.3 MHz, has been demonstrated for the detection of the gene sequence of *Staphylococcus aureus* (*S. aureus*) [309]. Device sensitivity improvement was achieved by using single layered graphene (SLG) with AuNP as the sensing area between the IDT, on top of the waveguide layer. AuNP served to increase the ssDNA probe binding density. Device exhibited an LOD of 1.86 pMol/L (12.4 pg/mL). The working of the DNA sensor with the SLG-AuNP sensing layer is shown schematically in Figure 34, with an increase in the sensor phase shift proportional to the increase in the concentration of the target.

(iv)Pathogenic Immunosensor: Virus and Bacterial Microorganism Detection

The rapid and real-time detection of pathogenic microorganisms is one of the most important requirements in disaster and medical emergencies, to improve disease control and patient management [312]. SAW biosensors provide a method to selectively and sensitively detect pathogenic microorganisms, such as HIV-1/HIV-2, Ebola virus in human serum, and foodborne and waterborne bacteria, such as *E. coli*, and biological warfare agents, such as bacteriophage M13, Coxsackie B4 virus, and Sin Nombre virus (SNV) [56,313,314,315,316].

A Love wave biosensor using 36° YX LiTaO_3_ with Al IDT and 500 nM SiO_2_ guiding layer operating at 325 MHz, was reported to detect the human immunodeficiency virus (HIV) with the capability to differentiate between the two serotypes (HIV-1 and HIV-2) from human serum [313]. A Lamb wave device utilizing a thin AlN film of 4.5 µm thickness on a flexible substrate of 125 µm thick polyethylene naphthalate (PEN) with Al IDT, operating at a Lamb wave frequency of 500 MHz, was demonstrated for the detection of *E. coli* [56,317]. The fabricated flexible device is given in Figure 35a. The device surface was utilized for the sensing, by immobilizing the antibody bioreceptors. The *E. coli* strain with concentration expressed in colony-forming units (CFU) was preincubated with the antibody specific to the target. The antibody-bound target, when introduced to the functionalized sensor surface, was adsorbed in the area in which the acoustic energy density was the highest, which was the IDT area. This is demonstrated in Figure 35b, with the IDT area showing the bacteria adsorption, at various concentrations. This resulted in an improvement in sensitivity, which came from the use of the whole sensor area as the active area. The phase response of transmission coefficient S_21_ of the Lamb wave device, showed a reduction in frequency upon the introduction of the sample target, which indicated mass loading, as shown in Figure 35c. The device showed an LOD of 6.54 × 10^5^ CFU/mL. More applications of SAW sensors in detecting pathogens can be found in Appendix A.

## 3. Acoustic Microfluidic Actuation

Apart from biosensing applications, piezoelectric acoustic waves are also applied in microfluidic actuation platforms used in LOC devices [30]. These applications include micromixers, micropumps, droplet generators, atomizers, cell separation and sorting, manipulation of cells and biomolecules, virus and bacteria separation, DNA/RNA separation, and cell lysis [318]. Some advantages that these devices possess are ease of fabrication, miniature-sized, biocompatible, non-contact, and a label-free technique [319,320]. The basic principle behind these devices comes from acoustic streaming force and acoustic radiation force.

The interaction of high amplitude acoustic oscillations with flowing liquid or droplets, results in the attenuation of acoustic energy and moment resulting in acoustic streaming. The acoustic streaming force is the working principle for fluid transport, micromixers, micropumps, droplet generators, and atomizers, which find application in LOC devices for disease diagnosis, drug development, and delivery, as well as in accelerating DNA analysis, which finds application in genomics [318]. Fluid mixing and transport play an important role in accelerating the biochemical reactions in LOC devices. The acoustic streaming force also minimizes the non-specific binding (NSB) in the LOC and POC devices, which helps to minimize the interference effects from non-target analytes.

The propagation of acoustic waves through a fluid, results in the generation of pressure waves, which induces acoustic radiation force (ARF) [321]. By controlling the power, frequency, and phase of the acoustic waves, as well as modifying the dimensions of IDT and microfluidic channels, the acoustic waves can be focused and can produce pressure nodes and antinodes, which helps in the trapping and manipulation of biological entities, such as cells and biomolecules, non-invasively. This forms the backbone of acoustophoresis, which utilizes acoustic waves to manipulate cells in a non-contact and label-free method [322]. ARF is the principle used in cell separation and sorting, cell manipulation, cell trapping, pathogen separation, and cell lysis, which finds application in disease diagnosis, molecular biology, and flow cytometry [323,324,325]. The use of ARF in cell separation finds several applications, such as the separation of circulating tumor cells (CTC), extraction of rare cells, as well as in the separation of white blood cells (WBC) from whole blood [325,326,327,328,329,330].

Piezoelectric bulk substrates, such as LiTaO_3_, LiNbO_3_, and quartz, and deposited thin piezoelectric films, such as AlN and ZnO, are generally used for generating acoustic waves for microfluidic actuation mechanisms. As these materials were covered in the previous sections, they will not be discussed in detail, here. Similar to biosensors, both BAWs and SAWs are employed for microfluidic actuation. As the SAW energy is concentrated on the surface of the piezoelectric materials, when compared to BAW, SAW-based microfluidic devices require less power to function [331]. The following section gives a brief introduction to the operating principle of acoustic wave-based microfluidic actuation with its application in biosensing.

### 3.1. BAW-Based Microfluidic Actuation: Operating Principle

The use of BAWs in microfluidic actuation devices has been reported for cell separation, sorting, and manipulation. The basic architecture of the device involves a piezoelectric bulk transducer that is incorporated at the bottom of a microfluidic channel, which acts as a resonator. The basic schematic of a BAW-based microfluidic device is shown in Figure 36a. The microfluidic channels can be fabricated from silicon, glass, or Al, which has a higher acoustic impedance [332]. This enables the formation of resonant chambers by reflecting the acoustic waves from the rigid sidewalls. The acoustic impedance mismatch between the fluid and the sidewalls of the microchannels results in the reflection of acoustic waves and forms standing waves in the fluid. Standing waves are formed inside the fluid-filled channel forming pressure nodes (point of minimum pressure) and antinodes (point o maximum pressure) when the transducer is excited by the AC power supply. An acoustic resonance mode is created across the channel width (w) with a resonant frequency given by $f=\frac{c_{c}n}{2w},$ with the acoustic velocity in the fluid represented by $c_{c}$ and n represents the order of the resonant mode [333]. They exert an acoustic radiation force on the particles leading to separation and trapping, which is given by [334]
(15)$$F_{\mathrm{rad}}{=4\pi \mathrm{ka}}^{3}E_{\mathrm{ac}}\emptyset \left(\beta ,\rho \right)\mathrm{Sin}\left(2\mathrm{kz}\right),$$
(16)$$\emptyset \left(\beta ,\rho \right)=\frac{1}{3}\left[\frac{5\rho -2}{2\rho +1}-\beta \right],$$
where $\emptyset \left(\beta ,\rho \right)$ is called the phase factor, which depends on the density and compressibility of the particle to be manipulated as well as the surrounding fluid, a is the radius of the particle, $k=\frac{2\pi }{\lambda }$ is the wavenumber, *λ* is the acoustic wavelength, and *z* is the axial distance from the pressure node [321]. $E_{ac}$ represents the acoustic energy density, $\rho =\frac{\rho_{p}}{\rho_{0}}$ is the ratio of density of the particle to the density of fluid, and $\beta =\frac{\beta_{p}}{\beta_{0}}$ is the ratio of the compressibility of the particle to the compressibility of the surrounding fluid. A schematic of standing waves formed inside the microchannel due to BAWs with the ARF acting on a particle, is shown in Figure 36b.

The phase factor, $\emptyset \left(\beta ,\rho \right)$ can be either positive or negative depending on the particle’s physical properties and controls the movement of the particle to either pressure node (+φ) or antinode (−φ). Generally, for cells, the phase factor is positive, and, hence, under the influence of ARF, they move towards the pressure node. By properly aligning the pressure nodes inside the channel width, efficient particle separation can be achieved by using BAW microfluidic devices. As shown in Equation (15), the ARF has a dependence on the particle radius, with a stronger force on bigger particles, thus enabling the separation of sub-micron particles from micron-sized particles [336]. Apart from the particle separation based on size, BAW-based devices have also been used to separate similar dimension particles, but with a different acoustic phase factor. The separation of PDMS particles from polystyrene particles, has been demonstrated by using a BAW transducer, which operates on the principle of difference in phase factor due to the difference in particle density and compressibility [335]. A schematic of acoustic separation of similar-sized PDMS, and polystyrene particles using BAWs, is shown in Figure 36c. The separation of particles using the BAW microfluidic device has also been reported, based on the particle size with acoustic streaming force and ARF [337].

#### 3.1.1. BAW Based Microfluidic Biosensing Applications

The separation, manipulation, and sorting of biological samples, such as cells, DNA, viruses, and bacteria, find application in clinical sample preparations. The separation of nano and submicron particles from micron-sized particles has been demonstrated by combining acoustophoresis with the fluid relocation principle [336]. The separation of *E. coli* from bovine red blood cells (B-RBC) has been demonstrated, using the above principle. The microchannel was made of Si, with a piezoelectric bulk transducer operating at 2.9 MHz mounted at the bottom of the channel to generate BAWs. The smaller *E. coli* from mixed samples were pulled towards the lateral stream via acoustic relocation, and bigger B-RBC remained at the central stream via acoustic radiation force, which resulted in the efficient separation of microorganisms from the blood sample.

Microbubble-based separation, sorting, and the manipulation of microparticles have also been widely used in LOC devices. Acoustically activated microbubbles, have been reported for manipulation of living animals, *Caenorhabditis Elegans* (*C. elegans*), which finds wide application in lab research for the study of various diseases [338]. The microfluidic device was formed on Al with a microchannel dimension of 3.1 mm × 4.2 mm × 0.5 mm, with microcavities being drilled with a dimension of 283 µm diameter and depth of 200 µm. The device schematic is shown in Figure 37a. The piezoelectric bulk transducer was bonded to one side of the channel to generate BAWs. The trapping and enrichment of *C. elegans* were performed by acoustic secondary radiation force around the acoustically actuated microbubbles, combined with the acoustic streaming induced drag force which attracted the worms to the microbubbles in the channel. The trapping of worms when the piezoelectric bulk transducer was turned on, is shown in Figure 37b,c.

The separation of CTC from white blood cells has also been demonstrated using a single inlet acoustophoretic microfluid chip, which has two stages for the pre-alignment and separation by acoustophoresis [339]. The microfluidic channel had two sections with a narrow first section for pre-alignment, which supported two pressure nodes from standing BAWs that trapped the particles in two dimensions. The aligned particles entered a wider second section of the microchannel for separation, with a single pressure node formed by the standing BAWs at the center. Larger particles in the separation zone experienced a stronger ARF, which pulled them to the channel center, while the smaller particles remained closer towards the channel sidewalls.

Acoustic standing wave-induced radiation force has also been reported for determining cell compressibility. Determining the physical properties of cells, such as compressibility, plays an important role in LOC devices for cancer disease diagnosis as the cancer cells exhibit more compressibility compared to normal cells [340]. A microfluidic device with Si microchannels made by DRIE, used along with a piezoelectric ceramic transducer mount at the bottom of the channel, was reported for determining the cell compressibility by utilizing ARF [341]. A mixture of polystyrene particles and cells was introduced into the channel. The known physical parameters of the polystyrene particles, such as the size, compressibility, and density, were used to calculate the acoustic energy density (E_ac_) inside the channel. Having calculated the E_ac_, the compressibility of the cells was calculated by tracing their trajectory inside the channel, with the assumption that both particles were subjected to the same acoustic energy inside the channel.

Apart from being used for cell separation and manipulation, BAW acoustophoresis using piezoelectric ceramic transducers has also been demonstrated for digital microfluidic applications, involving droplet merging, exchange of medium, and sorting [333]. Efficient integrated droplet mixing via acoustic streaming with a thin film AlN-based BAW device, has been reported for digital microfluidic platforms [342]. The inverse piezoelectric effect by an AlN-based BAW device, results in generating acoustic waves when electrically excited, which imparts an acoustic vortex in the droplets and performs the efficient mixing of samples.

### 3.2. SAW-Based Microfluidic Actuation: Operating Principle

SAW is widely used in acoustic microfluidic devices for cell sorting and separation, as well as for microfluidic fluid and droplet manipulation, such as micromixers, micropumps, atomizers, and fluid transport, which are used in LOC devices. SAW-based microfluidics has been widely used in both continuous channel devices with Si, glass, or PDMS channels, as well as with digital microfluidic platforms. The basic architecture of SAW-based microfluidic devices includes a SAW generation part, which includes IDT, and a fluid section that involves droplets (DMF) or the handling of fluids with microchannels (closed channel). There are two types of SAW waves used in microfluidic devices, namely traveling SAW (TSAW) and standing SAW (SSAW).

TSAW is produced by a single IDT on piezoelectric substrates, such as quartz, LiNbO_3_, or LiTaO_3,_ or on thin-film piezoelectric films, such as AlN/ZnO deposited on Si substrates. The IDT material used is, generally, a Ti/Au bilayer that offers biocompatibility. The device schematic of a TSAW based microfluidic device is shown in Figure 38a. Microparticle manipulation by TSAW is induced by either the acoustic streaming induced drag force, or by ARF due to pressure gradients, as schematically shown in Figure 38b. The acoustic radiation force induced on a microparticle of radius *R_p_* by TSAW is given by [321,328]
(17)$${F=2\pi \rho }_{l}{\left|A\right|}^{2}{{(\mathrm{kR}}_{P})}^{6}\frac{{9+2(1+\lambda }_{p}{)}^{2}}{{9(2+\lambda }_{p}{)}^{2}},$$
where *A* is the velocity potential’s complex amplitude, *k* is the wavenumber, $\lambda_{p}=\frac{\rho_{l}}{\rho_{p}}$ with $\rho_{l}$, and $\rho_{p}$ is the surrounding fluid and particle density. Equation (17) shows that the ARF has a stronger dependence on particle dimension. The manipulation force on particles is determined by a dimensionless parameter termed as the k-factor, which is given by $k=\frac{\pi d}{\lambda }$, where d is the diameter of the microparticle and λ is the acoustic wavelength in the fluid [321,343]. For *k* << 1, which is applicable for particles with dimensions much smaller than λ, the dominating force is the acoustic streaming force.

The acoustic streaming force is induced when high-frequency acoustic waves are attenuated by the fluid due to the fluid viscosity. As discussed in Section 2.2.1, the Rayleigh mode of SAW is utilized to induce acoustic streaming due to the coupling of acoustic energy into the surrounding medium. The acoustic waves undergo refraction when they come into contact with fluid due to a change in the acoustic velocity in the piezoelectric substrate and in the fluid. The angle of refraction, θ_R_, as indicated in Figure 38c, is given by [318]
(18)$$\theta_{R}{=\sin}^{-1}\frac{V_{L}}{V_{S}},$$
where $V_{L}$ and $V_{S}$ represent the acoustic velocity in the liquid and piezoelectric substrate, respectively. The acoustic streaming force has been used in several droplet manipulation schemes, such as droplet mixing, droplet splitting, transport, nebulization, and atomization [318,343]. Acoustic streaming-based SAW microfluidic devices find application in cell lysis for the release of DNA/RNA or microorganisms for biosensing [323,324]. For microparticles, such as cells and biomolecules with k > 1, the dominating force is ARF, which helps in particle trapping, separation, and sorting. A much smaller sorting area with improved accuracy can be achieved, by generating TSAW using a focused IDT (FIDT) [344]. A highly focused TSAW is generated with the sorting width of the order of tens of µm, which enables the sorting of a single microparticle.

SSAW is produced by two oppositely traveling SAW, generated by a parallel set of IDTs located on opposite sides of the microchannel on the face of the piezoelectric materials. The device schematic of the SSAW-based device is shown in Figure 38d. Two counterpropagating SAW interfere constructively, to produce standing waves that form pressure gradients in the fluid in the path. The SSAW formed by a parallel set of IDTs on the piezoelectric substrate, and traveling SAW generated by a single IDT within a PDMS microfluidic channel, is schematically shown in Figure 38b. The ARF induced by the SSAW, draws the particles towards pressure nodes or antinodes depending on the phase factor, similar to those shown in Equations (15) and (16), and explained in the BAW microfluidic devices. For cells, the positive phase factor results in the movement to the pressure nodes under the influence of ARF. SSAW is used in the sorting, aligning, focusing, and patterning of microparticles, which find application in cytometry, cell washing, as well as in the separation of particles [32,33,345]. By carefully positioning a pressure node inside the microchannel by adjusting the SAW resonant frequency, the separation of particles can be performed based on their size as well as their physical properties, such as density and compressibility. The schematic of microparticle separation using SSAW is shown in Figure 38e, in which the particles are attracted towards pressure nodes and antinodes within the microchannel depending on the particle physical properties, such as size or density or compressibility, which determines the phase factor. The separation between two consecutive pressure nodes is a half wavelength, which limits the separation distance in conventional SSAW microfluidic devices with a parallel set of IDTs to a quarter wavelength. To enhance the efficiency of separation, tilted angle SSAW (taSSAW) devices have been developed, with IDTs placed at an angle to the microchannel [325,329]. This has shown that the separation distance can be increased to about more than 10 times, compared to conventional SSAW devices.

Two orthogonally arranged sets of IDTs on a piezoelectric substrate, generating orthogonal SSAW with the same frequency, were also used in the patterning of microparticles using SSAW [36,348,349,350]. A series of nodes and antinodes will be generated in a 2-dimensional pattern, which can trap the microparticles and cells in either nodes or antinodes. The transportation of particles from one node to another was also demonstrated along with patterning, either by tuning the resonant frequency of the SAW to alter the position of the nodes/antinodes by using chirped IDT, or by slightly modulating the SAW without changing the resonant frequency. Tunable microparticle patterning was also demonstrated by using slanted-finger IDT [349]. With slanted IDT, the periodicity of the IDT varied along the length, which generated SSAW with a varying wavelength between a pair of electrodes. This helps in patterning microparticles with different properties.

#### 3.2.1. SAW-Based Microfluidic Biosensing Applications

SSAW- and TSAW-based microfluidic devices were extensively used in biosensing applications in the area of cell sorting, cell and microorganism separation, cell manipulation, and pattering. These devices make use of both acoustic radiation force developed by the standing waves and traveling waves combined with the acoustic streaming induced drag force. Droplet manipulation in DMF platforms, such as mixing, splitting, droplet dispensing, atomization, and nebulization, is also being widely performed using SAW-based devices utilizing acoustic streaming forces. Several excellent reviews have been published, which focus on the use of SAW in microfluidic devices.

SAW-based devices have been utilized for the cell lysis of RBC to detect malaria parasites and the breaking of exosomes to release RNA from them [323,324]. They utilize the TSAW-induced acoustic streaming force to perform lysis. IDTs formed on LiNbO_3_ substrates were used to generate Rayleigh SAW, which resulted in generating an acoustic vortex in the sample of exosomes that induced lysis to release RNA [324]. The usage of SAW devices in cell lysis must be properly monitored, in terms of operating frequency and power to reduce sample heating.

Both SSAW and TSAW, have been widely used for cell and microparticle separation. Detailed reviews on the use of SAW in microfluidic cell manipulations can be found elsewhere [321,343,346,351]. SSAW has been utilized to separate platelets from whole blood with 98% pure platelet extraction [352]. Separation of CTC from the patient sample has been demonstrated by utilizing taSSAW, as well as by using SSAW with a vertical resonator within the microchannel [326,353]. Sorting and separating CTC from red blood cells (RBC) in the flowing blood was demonstrated with 90% collection efficiency for glioma brain tumor cells, by utilizing a multistage SAW microfluidic device [328]. The device had two sets of Al IDTs on LiNbO_3_ substrate: one pair of IDTs for generating SSAW, which helped to focus the RBC and tumor cells in the first stage and a pair of focused IDTs (FIDT), of which one is excited for generating focused TSAW in the second stage. This helped to push the CTC away from the RBC. The device schematic with the fabricated device is shown in Figure 39a,b. The unidirectional acoustic radiation force by TSAW helped to focus the CTC in the direction of wave propagation and enhanced the efficiency of sorting. The separation and sorting trajectory of CTC from RBC is shown in Figure 39c.

Apart from cell separation, SAW-based microfluidic devices were also reported for the separation of bacteria from the blood, which helps to identify pathogens. SSAW microfluidic devices have been reported for the separation of *E. coli* from blood samples with a purity of 95% to 96% [327,354]. The label-free, non-invasive separation of exosomes from whole blood cells was demonstrated by using SSAW with a 99.999% of blood cell removal rate [355]. Exosomes play an important role in disease diagnosis and health monitoring. The device had a cell removal module and exosome isolation module with a separate set of IDTs. The cell separation module helped to isolate larger blood cells followed by the exosome isolation unit.

## 4. Future Perspectives

Acoustic biosensors and microfluidic devices are being continuously researched and developed due to the advantage of being non-invasive, label-free, and biocompatible. Acoustic-based biosensors utilizing BAW and SAW are a promising candidate for label-free biosensors. Being easy to fabricate, having fast device response, and requiring reduced sample volumes, these sensors can be integrated into LOC and POC devices. Even though QCM biosensors are readily available and operate in a liquid medium without acoustic damping, their lower sensitivity and the use of bulk quartz crystal substrate make them difficult to be integrated along with CMOS circuits for making a compact portable device. With the development of thin-film piezoelectric materials, the application of FBAR devices as biosensors opened up a new realm of devices that improved sensor sensitivity. The smaller dimension of the FBAR biosensors and the CMOS compatibility of thin piezoelectric films enables the fabrication of an integrated portable POC device. FBARs can also be fabricated in arrays due to their small dimensions, providing a higher throughput; however, FBAR-based biosensors are not commercially fabricated in large scales for POC devices. This can be attributed to the complex and expensive fabrication process involving the sputtering of thin films and the release of the thin membrane by backside cavity etch, which could only be performed in a cleanroom environment.

SAW biosensors are the most studied and developed acoustic biosensors due to their higher operating frequency in the range of 10 MHz to 500 MHz, providing higher device sensitivity. SAW energy, being concentrated on the substrate surface within a depth of a few acoustic wavelengths, makes them a low power consumption device compared to BAW-based biosensors and actuation platforms. SAW biosensors have a higher possibility for commercialization for low-cost LOC and POC, compared to QCM and FBAR. This is due to their ease of operation, wireless capability, user-friendly, and microfluidic channel integration capability coupled with their higher sensitivity. The integration of SAW biosensors with a miniature antenna on the piezoelectric quartz substrate, has been demonstrated for developing a wireless SAW sensor for the real-time continuous monitoring of blood pressure [356]. The SH-SAW sensor has also been demonstrated, with the integration of smartphone technology to develop a low-cost, rapid POC device for the detection of HIV [54]. A disposable, portable unit for monitoring blood hemostasis was demonstrated via the Love mode single port SAW resonator, using an LiNbO_3_ substrate [357]. Even though several demonstrations have been made for the successful application of SAW biosensors in the POC devices, the commercialization has not been conducted at a large scale. This can be attributed to the fabrication cost of SAW biosensors, which requires a cleanroom facility, the difficultly to integrate piezoelectric bulk substrates with CMOS, as well as the microfluidics integration. Advancements in IDT fabrication methods, such as 3D printing and the development of piezoelectric thin films in the SAW biosensor applications, can be considered as a step forward to overcome these challenges. The commercially available sam5 SAW biosensor, which is the most advanced and most commonly used, is by SAW instruments (Germany) [276]. The device is a Love wave sensor with five channels and operates as a label-free device. The sensor is capable of monitoring the target protein adhesion by the gravimetric principle, as well as capable of responding to the viscoelastic and conformational changes of the targets, which plays an important role in monitoring cell adhesions and the study of blood coagulation.

Acoustic microfluidic actuation utilizing BAW and SAW, has been demonstrated for a wide range of applications, such as the separation, sorting, and manipulation of complex biological samples. The use of SAW microfluidic devices, utilizing the Rayleigh mode, has been demonstrated for performing complex sample preparation steps, such as the mixing, splitting, dispensing, nebulization, and atomization of droplets in DMF platforms. SAW microfluidic devices play an important role in LOC and POC devices, with their lower power consumption capabilities. The successful commercialization of acoustic microfluidic devices is affected by factors, such as the low throughput, the need for bulky supporting equipment, such as a syringe for fluid dispensing and microscopic stages, a short operating lifetime due to residue aggregation, and the need for cleanroom facilities to fabricate the devices. The development of thin-film piezoelectric materials opens up a new method of integrating CMOS compatible thin-film piezoelectric actuators for BAW acoustophoretic devices instead of bulk transducers, which enables high volume manufacturing and miniature devices [358]. The utilization of materials, such as Al instead of Si or glass for the channel resonator in the BAW acoustophoretic device, has been demonstrated, which enables the elimination of cleanroom facilities for manufacturing and thus reduces the device cost [332].

Apart from this, thin-film piezoelectric materials can also be deposited on polymeric substrates, enabling them to form flexible, portable, and reusable biosensors, and microfluidic devices. The capability of integrating acoustic sensing and actuation in a single platform, due to the reversible property of piezoelectric materials, opens up the integration of entire sensing, manipulation, and sample preparation steps in a single miniature platform. An acoustofluidic device integrated with multimodal biosensors incorporating electrochemical and surface-enhanced Raman scattering immunosensors, has been reported for the diagnosis of Alzheimer’s disease (AD) [359]. A SAW-based microfluidic device with two separation zones has been efficiently used to separate AD biomarkers from human plasma with background contaminants. Acoustofluidic mixing enabled by PZT plates was also incorporated for the patterning of ZnO nanorod arrays for multimodal biosensing. Acoustofluidic-assisted bimodal biosensors have also been reported, which utilize SAW generated by using LiNbO_3_ substrates to focus the nanoparticles either at the center or perimeter of the glass capillary to excite the immunofluorescence assay or SERS [360]. Several challenges are yet to be addressed to realize this effort, as the SAW mode which excites actuation suffers significant signal attenuation while it is used for sensing. The demonstration of an integrated LOC device with microfluidics and biosensing capabilities on a single flexible platform has been reported by utilizing hybrid SAW modes [361]. The microfluidic droplet actuation steps of mixing, transport, and liquid dispensing have been demonstrated utilizing Lamb waves excited on ZnO film deposited on Al substrate with Cr/Au IDT. The biosensing capabilities were demonstrated on the same device by the excitation of thickness shear modes utilizing lateral field excitation. The sensing of anticancer drug Imatinib was reported by utilizing the aptamer as a bioreceptor coated on the ZnO film surface. Similar integrated sensing and droplet actuation has been reported on a digital microfluidic platform utilizing electrowetting on dielectric (EWOD) for actuation and FBAR sensor for biosensing [4]. The same piezoelectric AlN material has been shared as a dielectric for EWOD and as an active layer for a FBAR biosensor.

## 5. Summary and Conclusions

In this review, we provided an overview of the application of acoustic waves generated using piezoelectric materials in the field of biosensing and microfluidics. Acoustic wave biosensors and microfluidic actuation can be extended to multiple fields, such as clinical and medical research, the early diagnosis of deadly and infectious diseases, and environmental and food safety-related applications. Moreover, they are compact due to the smart electromechanical transduction property of piezoelectric materials, enabling the label-free, non-invasive, and biocompatible properties of the devices.

Two major categories of acoustic wave biosensors and microfluidic actuation platforms, namely BAW and SAW, have been reviewed in terms of the working principle and its applications. For BAW biosensors, two common types of devices are discussed. Firstly, the QCM biosensor, which has simpler fabrication procedures, excellent chemical and thermal stability, and ease of measurement data analysis. Next, the FBAR, which has the highest operating frequency among the acoustic wave biosensors in the range of 100 s of MHz to around 10 GHz, thereby providing the highest sensitivity. SAW biosensors are the most studied and developed acoustic biosensors, due to their higher operating frequency in the range of 10 MHz to 500 MHz without complex fabrication procedures, providing higher device sensitivity. The various sensing potentials of acoustic waves in myriads of applications, such as the sensing of proteins, disease biomarkers, DNA, pathogenic microorganisms, manipulation, and sorting of cells, have been reviewed. A summary of acoustic sensor performances in terms of advantages and limitations is listed in Table 3.

Microfluidic devices employ acoustic waves as actuation methods. Likewise, advantages, such as ease of fabrication, miniature sizes, biocompatibility, and non-contact and label-free devices, are also given here. The BAW has been demonstrated, here, to enable the trapping of cells and pathogens via acoustic radiation force and acoustophoresis. Cell compressibility diagnostics were also carried out using BAW actuation, which enables the determination of cancer cells from healthy cells. Microfluidic actuation using SAW was also discussed with the classification of SAW into TSAW and SSAW. Both TSAW and SSAW have been demonstrated for the manipulation and sorting of cells and microorganisms. TSAW makes use of ARF and acoustic streaming-induced drag force for particle manipulation, depending on the particle dimensions. SSAW makes use of ARF induced by the standing waves, which depend on the particle dimensions as well as the physical properties.

Herein, piezoelectric materials have proven a strong advantage by enabling both sensing and actuation within an integrated LOC and POC device due to the smart electromechanical transduction property. Studies in the field of thin-film piezoelectric material deposition methods and film property characterization can be considered as a step forward in the development of miniature acoustic wave biosensors. The real-time, label-free sensing capability of acoustic biosensors and microfluidic actuation platforms with sensitivity performance relevant to the clinical levels, makes them a suitable candidate for integrated devices enabling quick disease diagnosis as well as in the area of medical research. Especially in the current era of pandemic outbreaks, the label-free, real-time, and simple fabrication of piezoelectric acoustic wave biosensors enables them to be an excellent candidate for the identification of disease-causing viruses, as well as the disease biomarkers, such as antigens and antibodies. The advantage of less sample preparation and faster sensing capabilities, and the simplicity in the usage of piezoelectric biosensors and microfluidic platforms, enable them to be a promising candidate for rapid test kits for POC applications.

## Figures and Tables

**Figure 1 micromachines-13-00024-f001:**
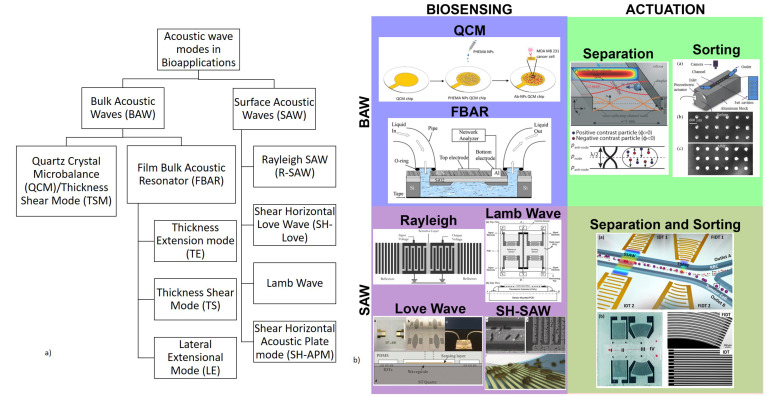
(**a**) Classification of acoustic wave modes used for bioapplications and (**b**) schematic representation of acoustic waves in biosensing and actuation application.

**Figure 2 micromachines-13-00024-f002:**
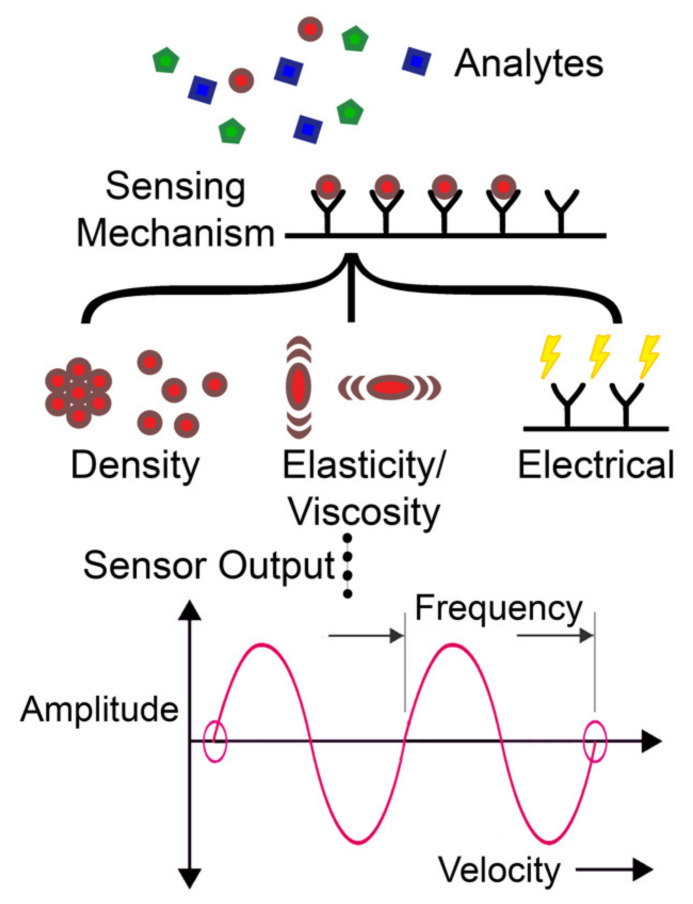
Basic working principle of acoustic biosensor showing the sensing mechanism. The sensor responds to the mass, viscosity, or electrical properties of the analytes which reflect as variation in the acoustic wave amplitude, phase/velocity, or frequency in the sensor output.

**Figure 3 micromachines-13-00024-f003:**
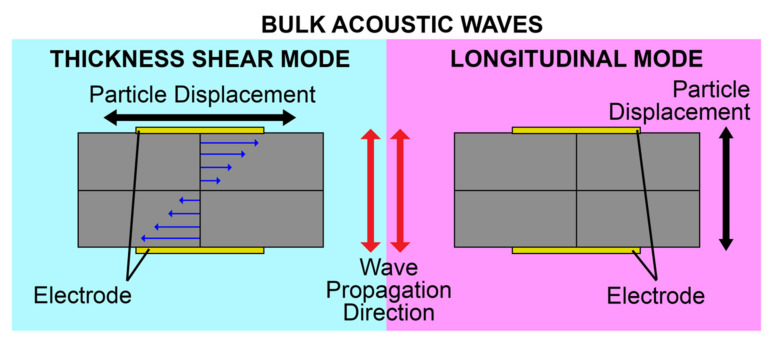
Modes of BAW. Thickness shear mode (TSM) and longitudinal mode.

**Figure 4 micromachines-13-00024-f004:**
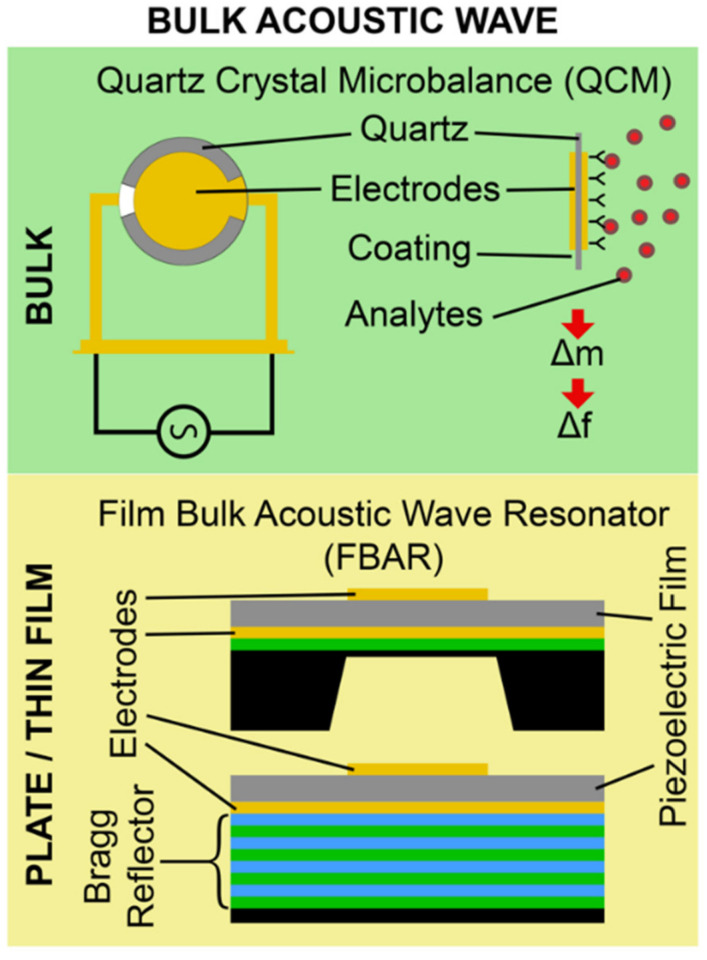
Classification of BAW devices used for biosensing.

**Figure 5 micromachines-13-00024-f005:**
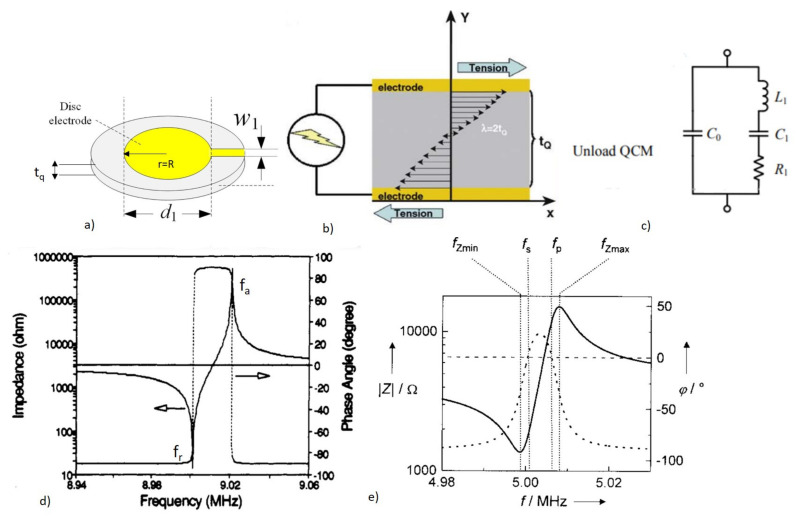
QCM architecture and basic principles. (**a**) Structural schematic of a QCM with a circular Au metallic electrode of radius R coated on the quartz crystal [64,65]. Reproduced with permission from Jianguo Hu et al., Electrochemistry Communications; published by Elsevier, 2020, Creative Commons (open access). (**b**) Acoustic wave propagation in the bulk with a particle displacement normal to the wave propagation direction on the application of the electric field [58]. Reproduced with permission from Guilherme N.M et al., Trends in Biotechnology; published by Elsevier, 2009. (**c**) BVD electrical equivalent circuit for unloaded QCM. C_1_, L_1,_ and R_1_ represent elastic stress, inertial mass, and damping in the resonator. C_0_ represents the static capacitance due to the top and bottom electrodes in QCM [66]. Reproduced with permission from Xianhe Huang et al., Sensors; published by MDPI, 2017 (open access). (**d**) The impedance response of unperturbed QCM showing both magnitude and phase response with the resonant frequencies f_a_ and f_r_. Reprinted (adapted) with permission from [67], copyright 1993, American Chemical Society. (**e**) Impedance magnitude and phase response for QCM in the presence of damping R_1_ ≠ 0, showing the splitting of f_r_ into f_s_, and f_zmin_ and f_a_ into f_p_ and f_zmax_ [41]. Reproduced with permission from Claudia Steinem et al., Angewandte Chemie International Edition; published by John Wiley and Sons, 2000.

**Figure 6 micromachines-13-00024-f006:**
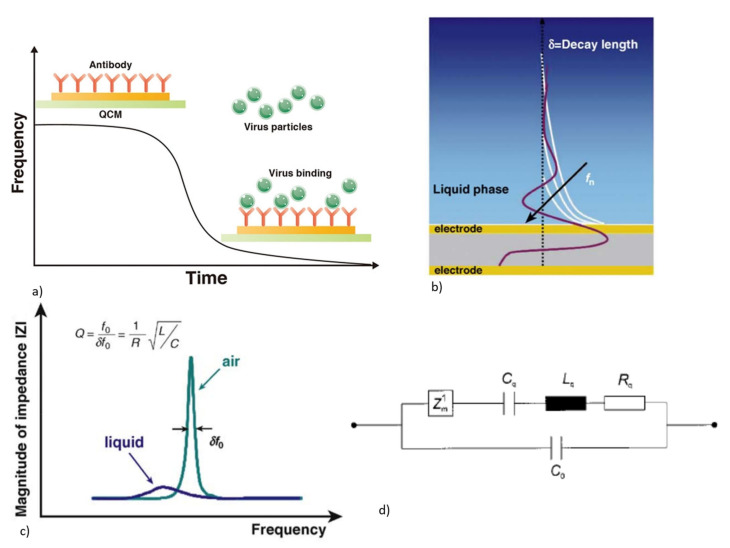
(**a**) Architecture of the QCM as a biosensor with the temporal variation in frequency [79]. Reproduced with permission from Constantinos Soutis et al., Advanced Materials; published by John Wiley and Sons, 2020, Creative commons (open access). (**b**) Performance of QCM in the liquid medium showing the shear wave penetration depth [58]. Reproduced with permission from Guilherme N.M et al., Trends in Biotechnology; published by Elsevier, 2009. (**c**)Variation in the Q factor of the QCM in the liquid medium. Reduction in frequency and reduction in amplitude and Q factor of the impedance response [58]. Reproduced with permission from Guilherme N.M et al., Trends in Biotechnology; published by Elsevier, 2009. (**d**) BVD equivalent circuit with loading impedance Z^1^_m_ added to the series motional arm [41]. Reproduced with permission from Claudia Steinem et al., Angewandte Chemie International Edition; published by John Wiley and Sons, 2000.

**Figure 7 micromachines-13-00024-f007:**
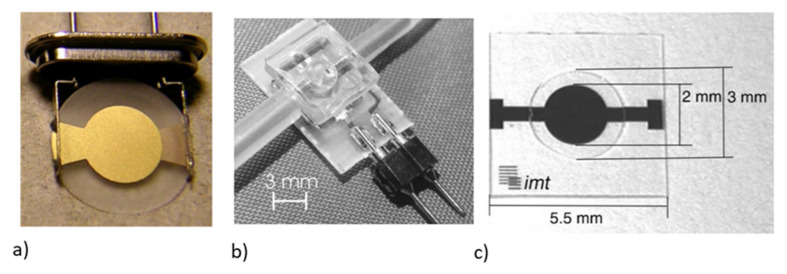
(**a**) Optical image of a standard QCM resonator with an Au electrode of diameter 4 mm [103]. Reproduced with permission from Henrik Anderson et al., Sensors and Actuators B: Chemical; published by Elsevier, 2007. (**b**) QCM integrated with a PDMS flow channel for sensing in liquid medium [105]. Reproduced with permission from Monika Michalzik et al., Sensors and Actuators B: Chemical published by Elsevier, 2005. (**c**) Optical image of an HFF QCM with an Au electrode with a thin quartz 3 mm diameter membrane [105]. Reproduced with permission from Monika Michalziket al., Sensors and Actuators B: Chemical; published by Elsevier, 2005.

**Figure 8 micromachines-13-00024-f008:**
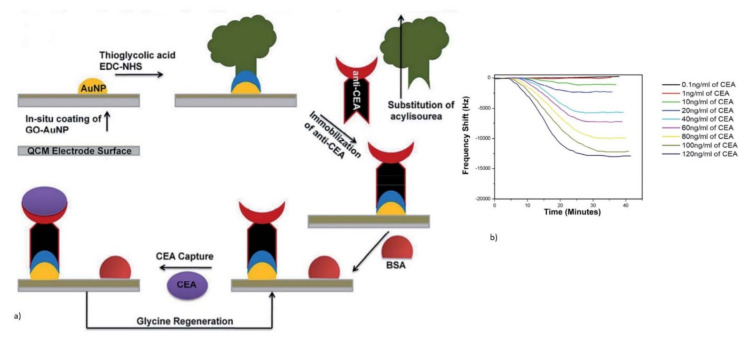
(**a**) Schematic of the surface functionalization of the anti-CEA antibody and sensing. (**b**) Increase in the resonant frequency shift with the CEA concentration [112]. Reproduced with permission from P. J. Jandas et al. [112], copyright 2020, RSC Advances; published by The Royal Society of Chemistry, Creative Commons (open access).

**Figure 9 micromachines-13-00024-f009:**
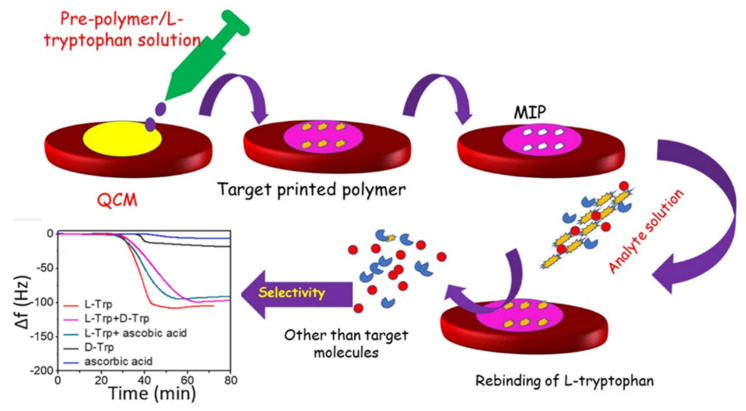
QCM biosensor coated with the MIP for L-Tryptophan detection, with high selectivity to the target from the complex medium. The selectivity to similar molecules, such as D-Trp and ascorbic acid, is shown as no response on the frequency shift [129]. Reproduced with permission from K. Prabakaran et al., Colloids and Surfaces A: Physicochemical and Engineering Aspects; published by Elsevier, 2021.

**Figure 10 micromachines-13-00024-f010:**
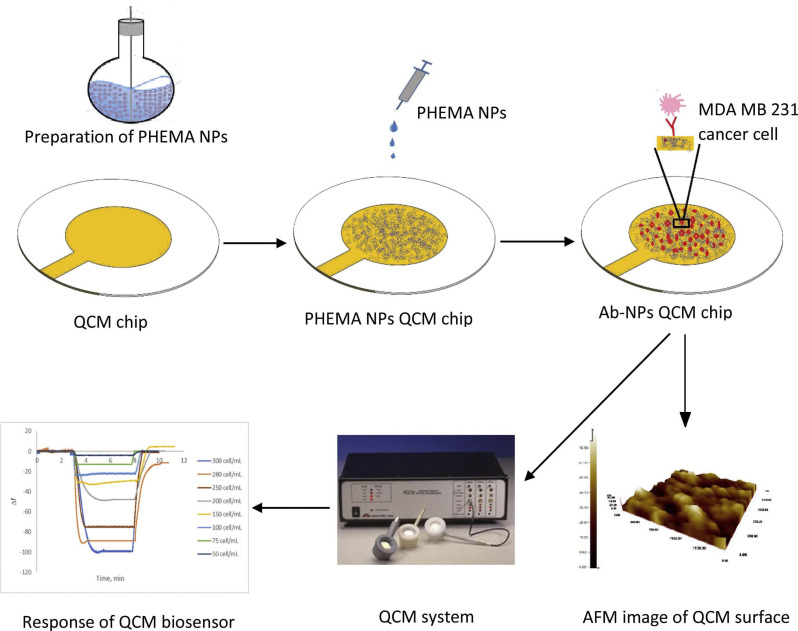
QCM biosensor for the detection of the metastatic potential of MDA MB231 human breast cancer cells with surface modification using PHEMA-NPs to improve the bioreceptor density. Sensor response to varying target concentration shows the capability to quantitatively detect the cells [114]. Reproduced with permission from Monireh Bakhshpour et al., Talanta; published by Elsevier, 2019.

**Figure 11 micromachines-13-00024-f011:**
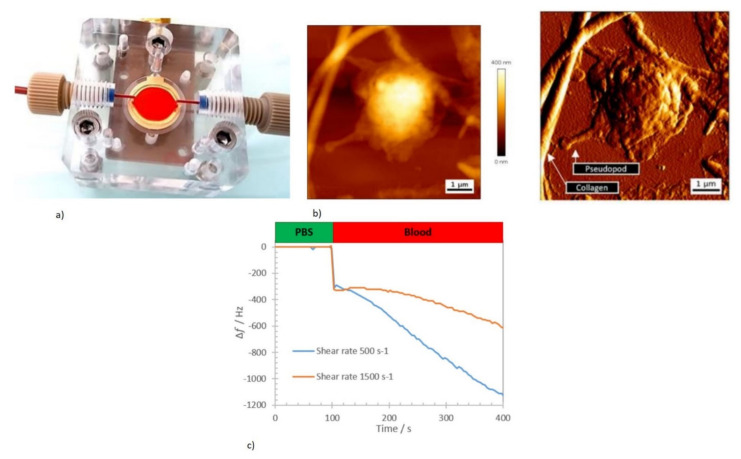
(**a**)Blood perfusion chamber with a QCM biosensor. (**b**) AFM image of platelet deposit on the collagen-coated QCM sensor surface. (**c**) Real-time monitoring of frequency shift at various wall shear rates to imitate the physiological blood microcirculation condition in the human body, showing the interaction of the platelet with collagen [144]. Reproduced with permission from Oseev A et al., Nanomaterials; published by MDPI, 2020 (open access).

**Figure 12 micromachines-13-00024-f012:**
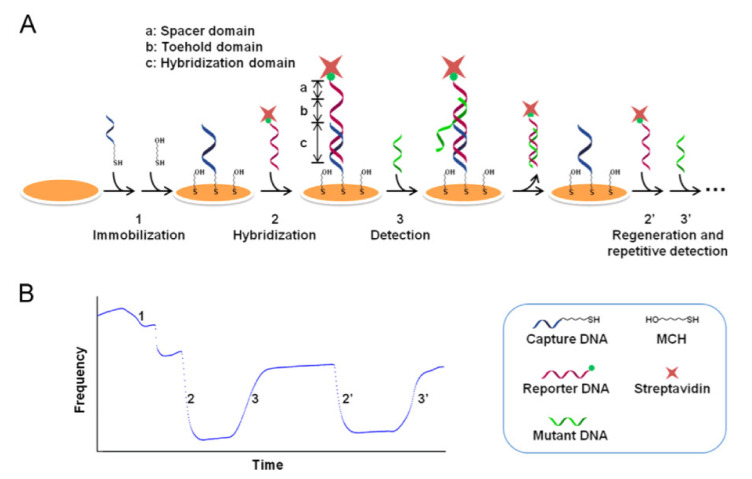
Schematic of the QCM DNA sensor for the successive detection of the single-base mutation and regeneration with a reporter DNA is shown in (**A**). Successive detection and regeneration achieved by using the SDR, as shown in real-time frequency response in (**B**), with 2′ representing the repetitive detection of the target mutant gene, and 3′ is the regeneration [155]. Reproduced with permission from Dingzhong Wang et al., Biosensors and Bioelectronics; published by Elsevier, 2013.

**Figure 13 micromachines-13-00024-f013:**
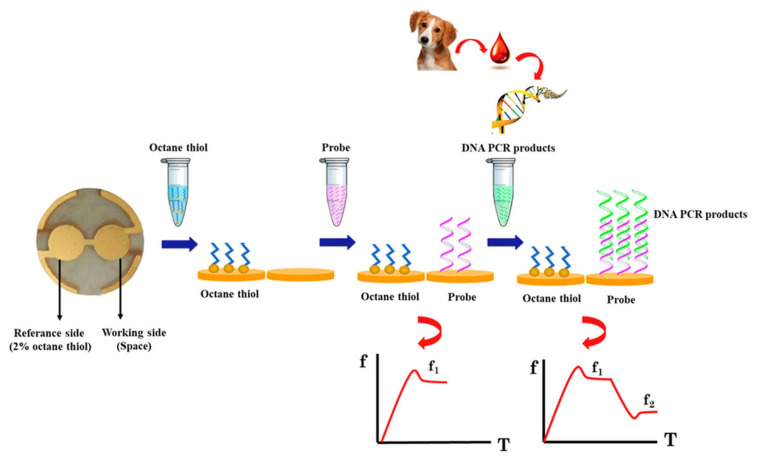
Dual electrode QCM for *Ehrlichia canis* in canines with PCR amplified DNA samples from blood and the corresponding frequency shift due to target–probe hybridization [156]. Reproduced with permission from Kespunyavee Bunroddith et al., Analytica Chimica Acta; published by Elsevier, 2018.

**Figure 14 micromachines-13-00024-f014:**
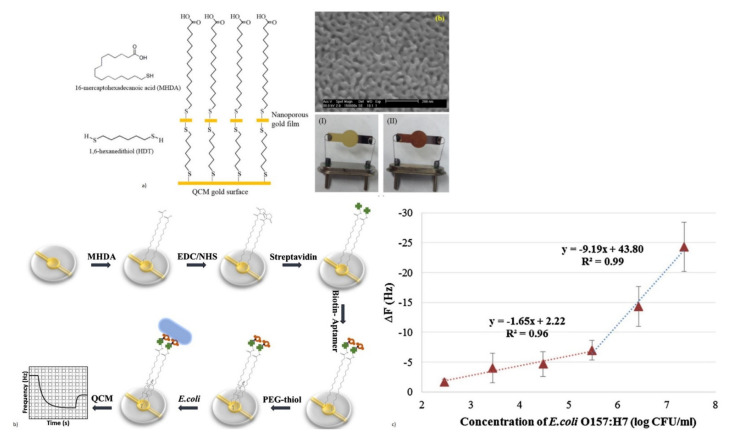
(**a**) QCM sensor with a nano well-structured top electrode for H5N1 detection. A nanoporous Au film is shown with an SEM image of the nanoporous film and an optical image of the Au electrode, before and after the formation of a nano well [118]. Reproduced with permission from Ronghui Wang et al., Sensors and Actuators B: Chemical; published by Elsevier, 2017. (**b**) QCM sensor functionalization for binding the aptamer specific to *E. coli*. (**c**) Aptasensor response to varying concentrations of target *E. coli* bacteria [166]. Reproduced with permission from Xiaofan Yu et al., Journal of Biotechnology; published by Elsevier, 2018.

**Figure 15 micromachines-13-00024-f015:**
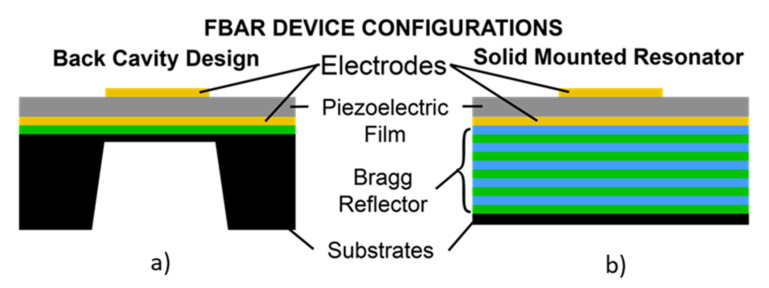
FBAR device configurations: (**a**) Back cavity design and (**b**) Solidly Mounted Resonator.

**Figure 16 micromachines-13-00024-f016:**
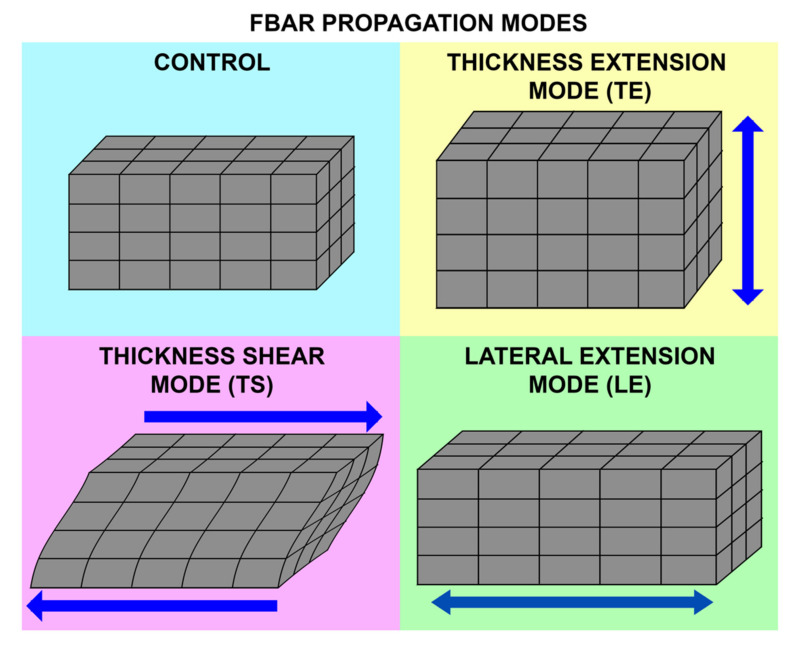
Propagation modes of FBAR based on the particle displacement.

**Figure 17 micromachines-13-00024-f017:**
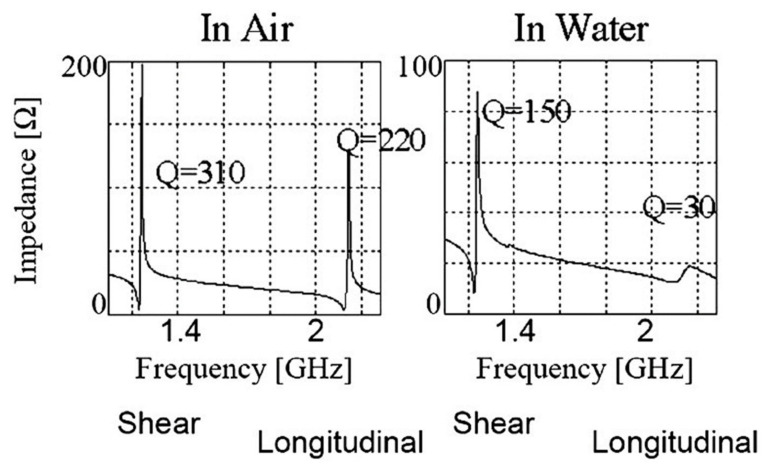
Q factor of longitudinal and shear mode in air and water, with significant damping of longitudinal mode in water [59]. Reproduced with permission from G. Wingqvist, Surface and Coatings Technology; published by Elsevier, 2010.

**Figure 18 micromachines-13-00024-f018:**
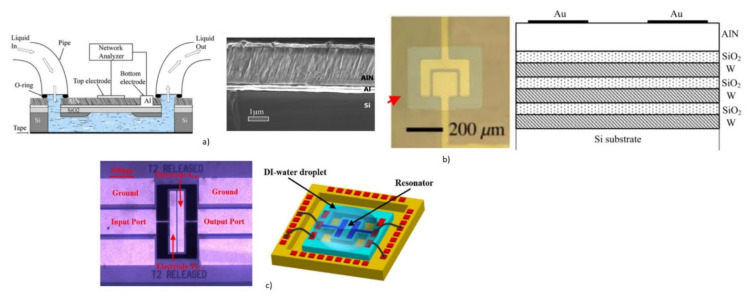
(**a**) Schematic of the TS mode FBAR with c-axis tilted AlN film with the SEM image of AlN showing the tilted angle [175]. Reproduced with permission from I. Katardjiev et al., Vacuum; published by Elsevier, 2012. (**b**) Optical image and schematic of the TS mode FBAR with lateral filed excitation on the c-axis oriented piezo film [49]. Reproduced with permission from Da Chen et al., Biosensors and Bioelectronics; published by Elsevier, 2017. (**c**) Optical image and schematic of the LE or contour mode FBAR sensor for the liquid medium [177]. Reproduced with permission from A. Ali et al., Sensors and Actuators A: Physical; published by Elsevier, 2016.

**Figure 19 micromachines-13-00024-f019:**
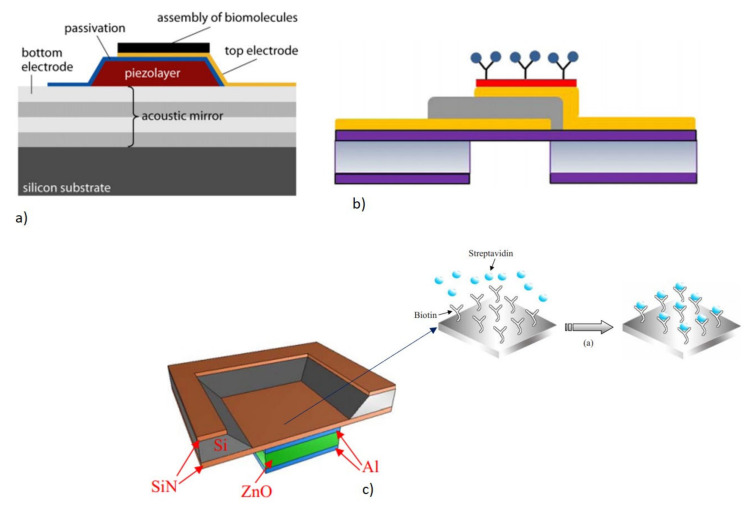
Various bioreceptor immobilizations for the different configurations of FBAR biosensors. (**a**) SMR with the top electrode as an active sensing area for bioreceptors [182]. Reproduced with permission from Martin Nirschl et al., Sensors and Actuators A: Physical; published by Elsevier, 2009. (**b**) Back cavity FBAR with the top electrode for the immobilization of bioreceptors [183]. Reproduced with permission from Tae Yong Lee et al., Thin Solid Films; published by Elsevier, 2010. (**c**) Back cavity FBAR with the backside Si3N4 membrane in the cavity coated with Au and used for the immobilization of bioreceptors [184]. Reproduced with permission from Hao Zhang et al., Applied Physics Letters; published by AIP Publishing, 2010.

**Figure 20 micromachines-13-00024-f020:**
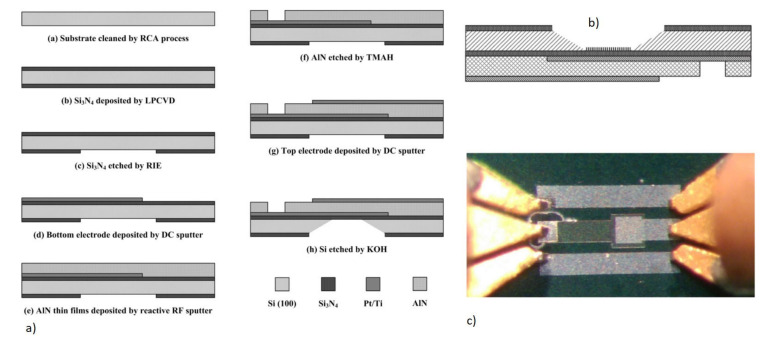
Fabrication of the FBAR biosensor. (**a**) Process flow in the fabrication of the FBAR device (**b**) FBAR biosensor with the back cavity functionalized using Cr/Au (**c**) Optical image of the fabricated device showing front side. Reproduced with permission from Ying-Chung Chen et al. [190], copyright 2015, Nanoscale Research Letters; published by Springer Nature, Creative Commons (open access).

**Figure 21 micromachines-13-00024-f021:**
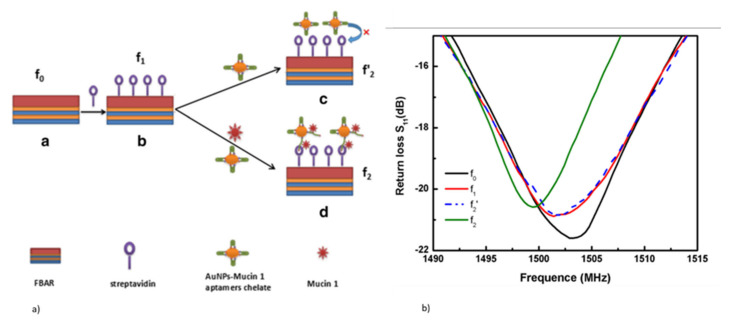
(**a**) Indirect MUC-1 detection scheme with streptavidin-coated on the FBAR surface and biotin bound aptamer. (**b**) Resonant frequency shift on the return loss spectrum for each layer of mass added corresponding to streptavidin binding and MUC-1 bound biotinylated aptamer. Reproduced with permission from Dan Zheng et al. [200], copyright 2016, Nanoscale Research Letters; published by Springer Nature, Creative Commons (open access).

**Figure 22 micromachines-13-00024-f022:**
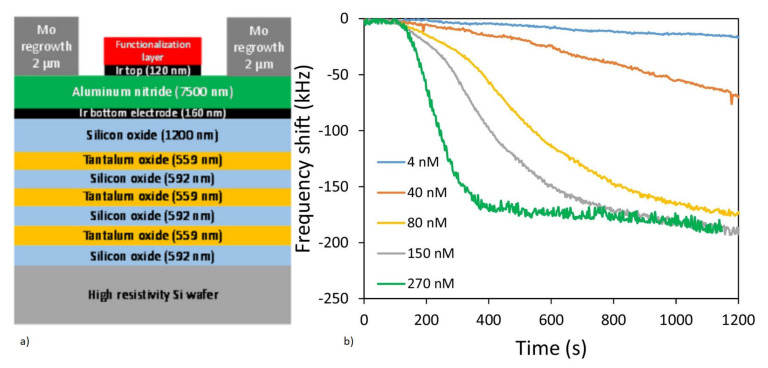
(**a**) Illustration of gravimetric SMR for operating in a liquid environment c-axis tilted AlN with Ir top and bottom electrodes. (**b**) Biosensing capability of the device demonstrated with the detection of varying concentrations of thrombin, by using a specific aptamer as a bioreceptor, with a reduction in frequency increasing with increase in target thrombin concentration due to mass loading [46]. Reproduced with permission from Mario DeMiguel-Ramos et al., Sensors and Actuators B: Chemical; published by Elsevier, 2017.

**Figure 23 micromachines-13-00024-f023:**
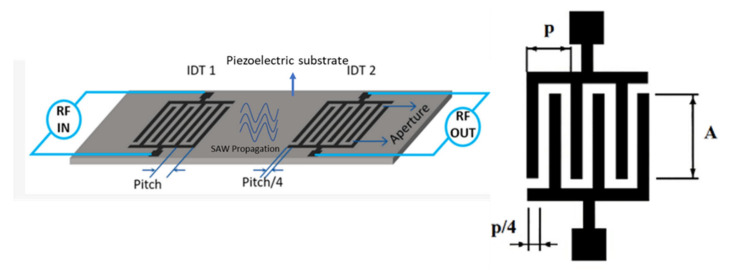
SAW device fundamentals with IDT on a piezoelectric substrate. Reproduced with permission from J. Devkota et al. [217], Sensors; published by MDPI, 2017 (open access). IDT pitch is represented as p and IDT aperture is given by A. Reproduced with permission from Rocha-Gaso et al. [209], Sensors; published by MDPI, 2009 (open access).

**Figure 24 micromachines-13-00024-f024:**
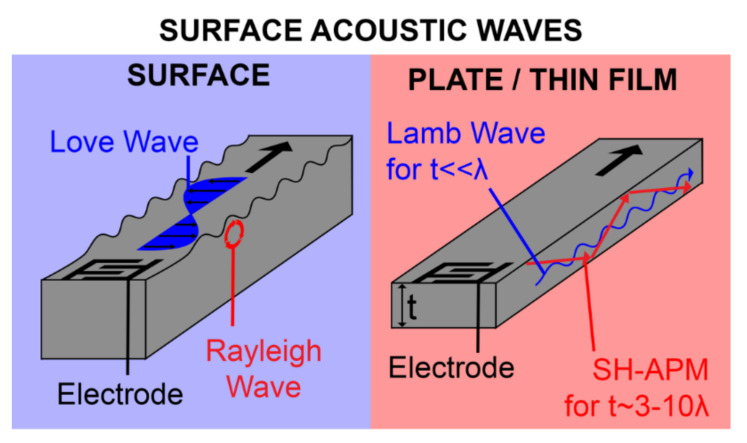
Propagation modes in surface acoustic waves.

**Figure 25 micromachines-13-00024-f025:**
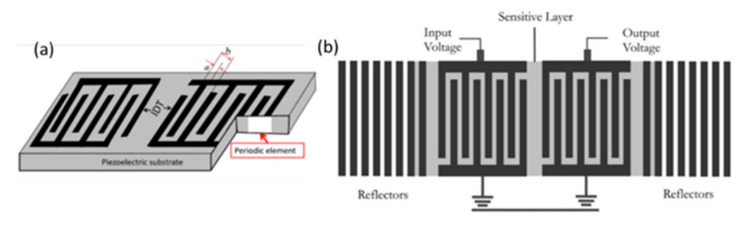
(**a**) Delay line configuration [203]. Reproduced with permission from Zhaoming Zhou et al., International Journal of Distributed Sensor Networks; published by SAGE, 2019, (open access) (**b**) SAW resonator configuration [210]. Reproduced with permission from Adnan Mujahid et al., Molecularly Imprinted Sensors; published by Elsevier, 2012.

**Figure 26 micromachines-13-00024-f026:**
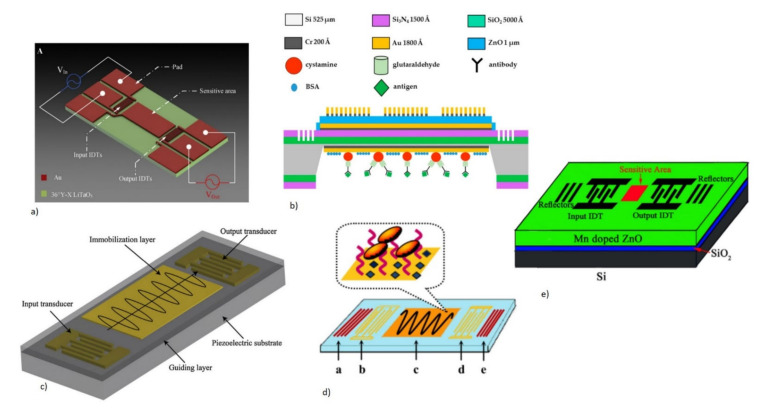
Schematics of various SAW mode biosensors. (**a**) SH-SAW biosensor with an Au IDT and sensing layer in the delay line configuration on bulk LiTaO_3_ [214]. Reproduced with permission from Junwang Ji et al., Sensors and Actuators B: Chemical; published by Elsevier, 2019. (**b**) FPW SAW biosensor with ZnO thin film piezoelectric layer. Functionalization of the sensor for biosensing is performed at the backside cavity. Reproduced with permission from J.-W. Lan et al. [261], Sensors; published by MDPI, 2016 (open access). (**c**) Love wave biosensor on a quartz substrate with SiO_2_ guiding layer and Au sensing layer on top of the guide in between the IDT [262]. Reproduced with permission from F. Di Pietrantonio et al., Sensors and Actuators B: Chemical; published by Elsevier, 2016. (**d**) SAW resonator for biosensing on LiTaO_3_ with an Au sensing layer in between the IDT and reflectors [263]. Reproduced with permission from Kai Chang et al., Biosensors and Bioelectronics; published by Elsevier, 2014. (**e**) SAW resonator biosensor on a sputter ZnO thin film [211]. Reproduced with permission from Jingting Luo et al., Biosensors and Bioelectronics; published by Elsevier, 2013.

**Figure 27 micromachines-13-00024-f027:**
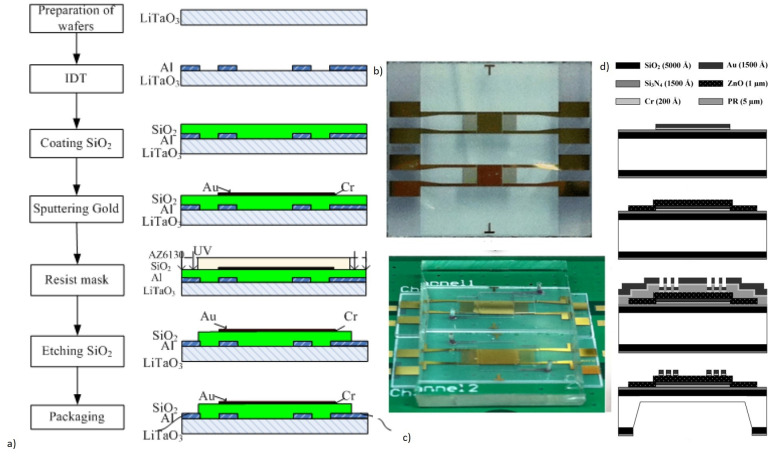
(**a**) Fabrication steps involved in Love wave biosensor. Reproduced with permission from F. Zhang et al. [272], Sensors; published by MDPI, 2015 (open access). (**b**) Fabricated Love wave device with an Au electrode and sensing area and SiO_2_ guide layer on quartz with a reference channel [273]. Reproduced with permission from Xi Zhang et al., Sensors and Actuators B: Chemical; published by Elsevier, 2015. (**c**) SAW biosensor integrated with PDMS microfluidic channels [271]. Reproduced with permission from Xi Zhang et al., Sensors and Actuators B: Chemical; published by Elsevier, 2014. (**d**) Fabrication steps for an FPW biosensor with a ZnO piezoelectric layer [247,274]. Reproduced with permission from I.-Y. Huang et al. [247], Sensors and Actuators B: Chemical; published by Elsevier, 2008.

**Figure 28 micromachines-13-00024-f028:**
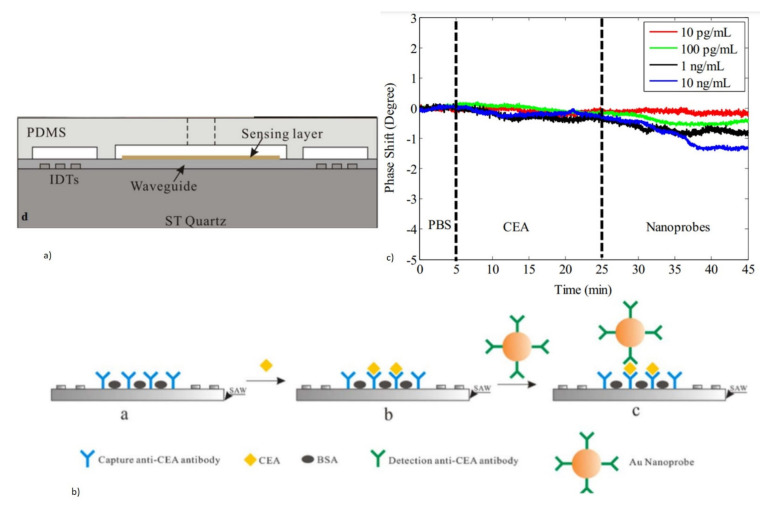
(**a**) SH-SAW Love wave sensor schematic with a PDMS microchannel for CEA detection. (**b**) Sandwich immunoassay scheme with an AuNP–detection antibody conjugate to enhance the mass loading. (**c**) Sensor phase shift response showing the enhanced output with the addition of nanoprobes [258]. Reproduced with permission from Shuangming Li et al., Biosensors and Bioelectronics; published by Elsevier, 2017.

**Figure 29 micromachines-13-00024-f029:**
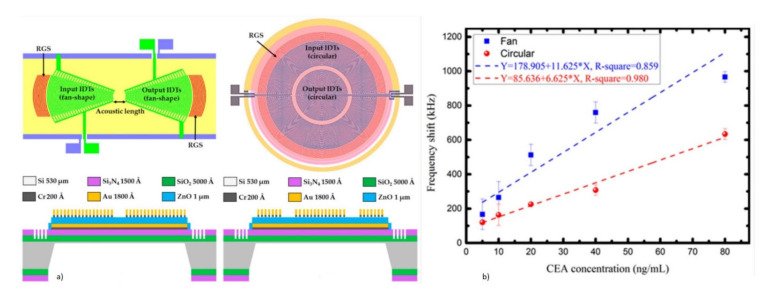
FPW device for the detection of CEA. (**a**) Schematic and cross-section of the fan-shaped and circular electrode. (**b**) Device frequency shift with an increase in the CEA concentration due to mass loading. Reproduced with permission from J.-W. Lan et al. [261], Sensors; published by MDPI, 2016 (open access).

**Figure 30 micromachines-13-00024-f030:**
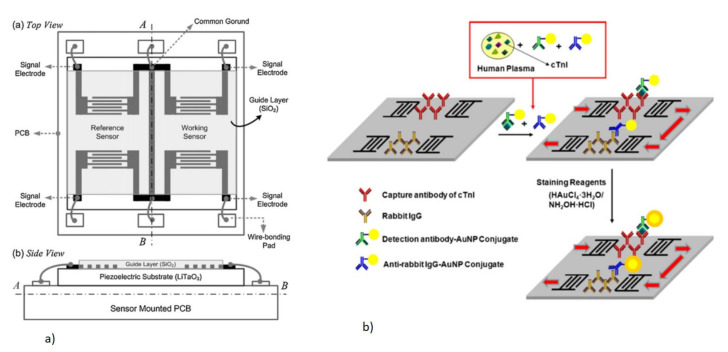
(**a**) Device schematic with reference and sensor channel with top view and side view of the Love wave sensor for cTnI detection. (**b**) Sandwich immunoassay schematic showing the capture of cTnI and Au staining signal amplification [281]. Reproduced with permission from Joonhyung Lee et al., Sensors and Actuators B: Chemical; published by Elsevier, 2013.

**Figure 31 micromachines-13-00024-f031:**
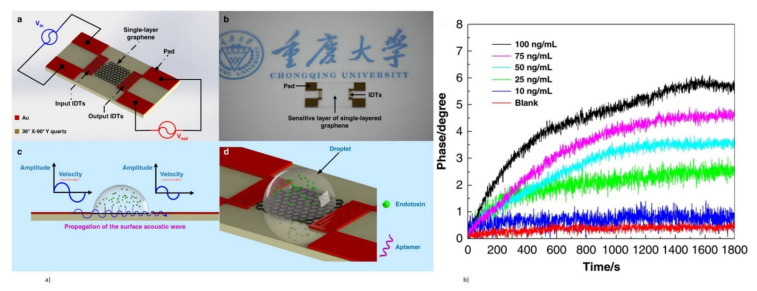
(**a**) Schematic of the SH-SAW device with SLG for sensing endotoxin with the fabricated device and SAW propagation perturbation by the endotoxin droplet. (**b**) Real-time change in phase of the device with increasing concentration of endotoxin. Reproduced with permission from Junwang Ji et al. [55], copyright 2020, Microsystems and Nanoengineering; published by Springer Nature, Creative Commons (open access).

**Figure 32 micromachines-13-00024-f032:**
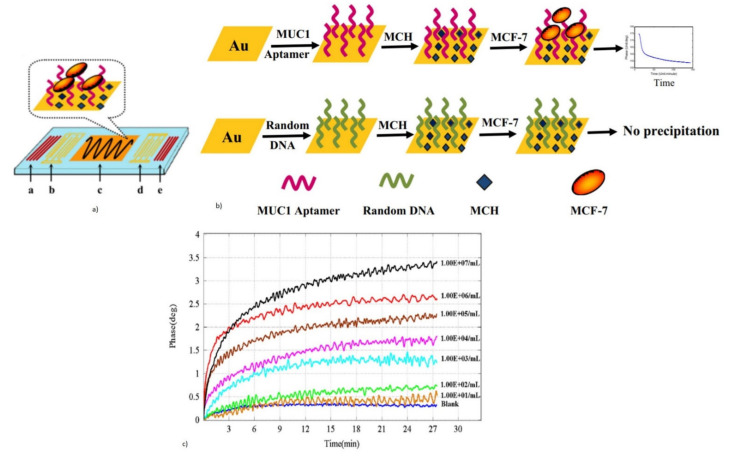
(**a**) Device schematic of a leaky SAW resonator with an Au sensing area in between IDT and reflectors. (**b**) Surface functionalization of Au sensing area with MUC1 aptamer for selective and specific detection of MCF-7 cells. (**c**) Real-time monitoring of device phase shift corresponding to varying concentrations of MCF-7 cancer cells [263]. Reproduced with permission from Kai Chang et al., Biosensors and Bioelectronics; published by Elsevier, 2014.

**Figure 33 micromachines-13-00024-f033:**
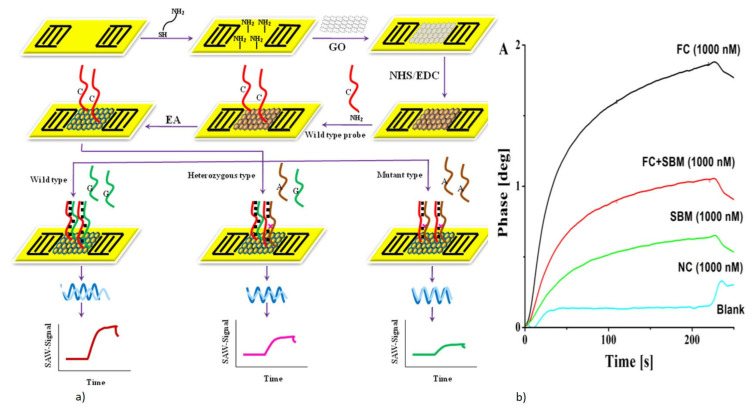
(**a**) Schematic of the SAW sensor for functionalization with GO and the capture probe DNA with variation in the phase signal for different mismatch target sequences for SNP detection in CYP2D6*10. (**b**) Variation in real-time phase monitoring of SAW device with FC, SBM, and FC+SBM mix, and the NC target sequence showing selectivity and sensitivity of the sensor. Reprinted (adapted) with permission from [307], copyright 2015 American Chemical Society.

**Figure 34 micromachines-13-00024-f034:**
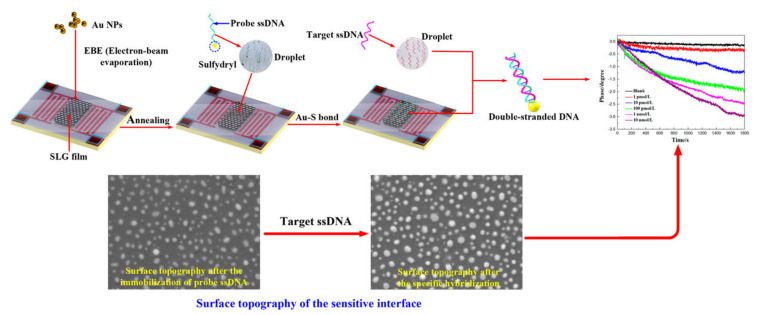
SH-SAW DNA sensor with SLG for the detection of the *S. aureus* gene sequence. Real-time monitoring of phase shows a response to an increase in the concentration of *S. aureus*. Reprinted (adapted) with permission from [309], copyright 2020 American Chemical Society.

**Figure 35 micromachines-13-00024-f035:**
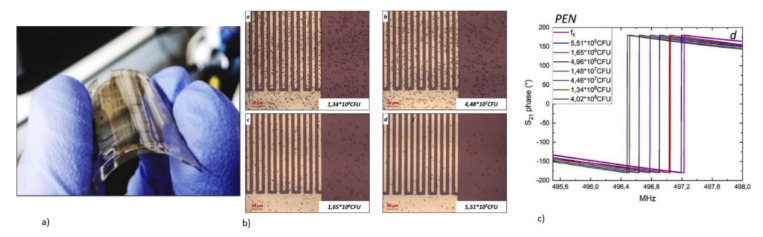
(**a**) Fabricated flexible biosensor on PEN with AlN and Al electrode for better acoustic matching and reduced loss. (**b**) Surface adsorption of *E. coli* on the sensor surface at the IDT area. (**c**) Sensor frequency redshift with an increase in the number of bacteria [56]. Reproduced with permission from Leonardo Lamanna et al., Biosensors and Bioelectronics; published by Elsevier, 2020.

**Figure 36 micromachines-13-00024-f036:**
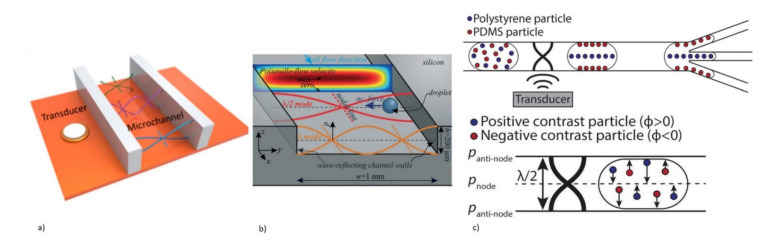
BAW microfluidic devices. (**a**) Basic architecture of a BAW-based microfluidic actuation device with a piezoelectric transducer and microchannel. Generated bulk acoustic waves produce standing waves inside the channel which serves as a resonator [321]. Reproduced with permission from Y. Gao et al., Micromachines; published by MDPI, 2020 (open access) (**b**) Schematic of standing waves with pressure nodes and antinodes inside the channel. ARF attracts the particles towards pressure nodes and antinodes depending on the phase factor. Reproduced with permission from Ivo Leibacher et al. [333], copyright 2015, The Royal Society of Chemistry; published by Lab Chip, Creative Commons (open access). (**c**) BAW-based microfluidic separation of particles based on particle physical property [335]. Reproduced with permission from Anna Fornell et al., Applied Physics Letters; published by AIP Publishing, 2018.

**Figure 37 micromachines-13-00024-f037:**
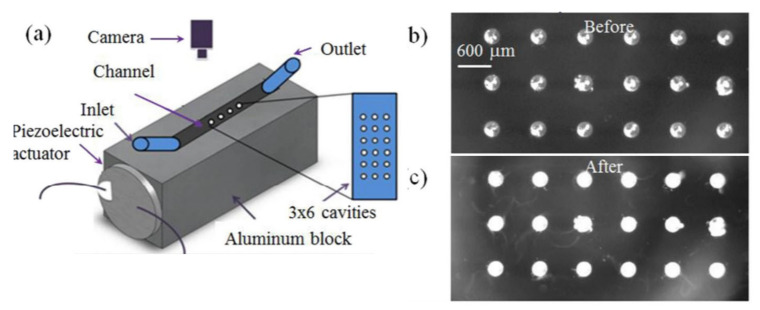
(**a**) Device schematic for microbubble-based acoustic separation of *C. elegans*. (**b**) Microbubble formed and no worm trapped before the acoustic excitation is turned on. (**c**) Trapping of worms towards the bubble when the acoustic transducer is turned on [338]. Reproduced with permission from Yuhao Xu et al., Applied Physics Letters; published by AIP Publishing, 2013.

**Figure 38 micromachines-13-00024-f038:**
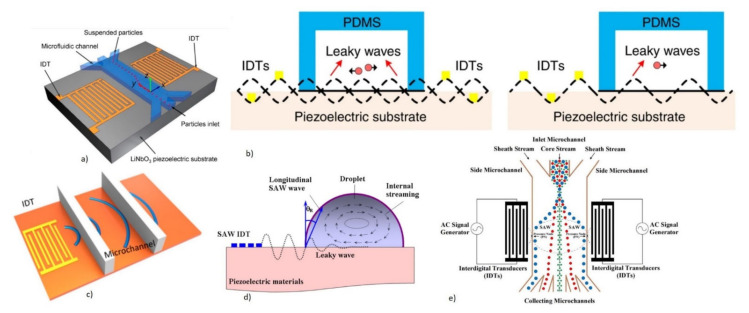
(**a**) Schematic of TSAW microfluidic device with a single IDT [321]. Reproduced with permission from Y. Gao et al., Micromachines; published by MDPI, 2020 (open access). (**b**) SSAW pressure nodes and antinodes formed within a PDMS microchannel and TSAW waves generated by a single IDT. Reproduced with permission from M. Wu et al. [346], copyright 2019, Microsystems and Nanoengineering; published by nature, Creative Commons (open access). (**c**) Acoustic streaming force generated by TSAW with SAW being refracted when in contact with fluid due to acoustic velocity difference. Reproduced with permission from J. K. Luo et al. [318], IntechOpen, copyright 2013, Creative Commons (open access) (**d**) Schematic of SSAW microfluidic device with parallel IDTs [347]. Reproduced with permission from Hsu et al., Micromachines; published by MDPI, 2019 (open access). (**e**) Schematic of SSAW-based device used for microparticle separation [330]. Reproduced with permission from Soliman et al., Bioengineering; published by MDPI, 2017 (open access).

**Figure 39 micromachines-13-00024-f039:**
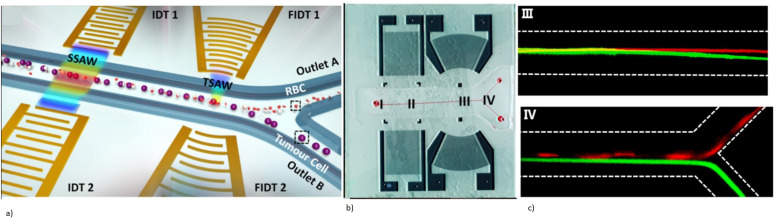
(**a**) SAW-based multistage microfluidic device for CTC separation from RBC. (**b**) Fabricated device with IDT and FIDT. (**c**) CTC separation trajectory in the positions marked III and IV in (**b**). The unidirectional TSAW generated by FIDT in III, pushes the CTC away from RBC and isolates them to outlets A and B in the section marked IV [328]. Reproduced with permission from Kaiyue Wang et al., Sensors and Actuators B: Chemical; published by Elsevier, 2018.

**Table 1 micromachines-13-00024-t001:** Piezoelectric properties of the substrate and thin-film materials used in acoustic biosensing.

Substrate	Electromechanical Coupling (%)	Acoustic Wave Velocity (m/s)	TCF (p pM/°C)	Dielectric Constant
64° YX LiNbO_3_	11.0–11.5	4330–4742	70–81	85.2
36° YX LiTaO_3_	5.0–7.6	4100–4212	28–35	43
X-112° Y cut LiTaO_3_		3300	18	43
ST-X Cut Quartz	0.0016	3159	0	3.8
Langasite (La_3_Ga_5_SiO_14)_	0.8	2734	0	18.23
ZnO	1.5–1.7	6336	−40~−60	8.66
AlN	3.1–8	11,050	19	8.5–10
PZT	20–35	4500	-	380
PVDF	2.9	2600	-	6–8

**Table 2 micromachines-13-00024-t002:** SAW modes and characteristics and application in biosensing. Pictures from [209,236]. Reprinted (adapted) with permission from Jay W. Grate et al. [236], copyright 1993, American Chemical Society.

SAW Mode	Device Configuration and Particle Displacement Profile	Substrate	Guide Layer	Operating Frequency	Substrate Thickness	Application
R-SAW	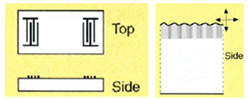	Quartz, LiNbO_3_, ZnO	NA	30–300 MHz	>>λ	Gas sensing
SH-SAW- Love Mode	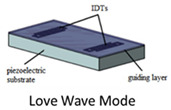	36° YX LiTaO_3_, ST-Quartz, 64° YX LiNbO_3_	SiO2, ZnO, PMMA, Photoresist	30–500 MHz	>>λ	Gas sensing, liquid sensing
Lamb Wave (FPW)	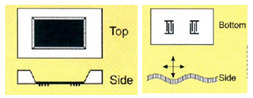	ZnO, AlN, PZT	NA	2–20 MHz	<<λ	Gas sensing, liquid sensing
SH-A PM	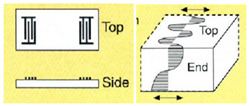	ST-Quartz ZX LiNbO_3_	NA	25–200 MHz	3–10 λ	Gas sensing, liquid sensing

**Table 3 micromachines-13-00024-t003:** Summary table for the comparison of acoustic sensors with the advantages and limitations.

Factors	BAW		SAW
	QCM	FBAR	
Advantages	Easy to fabricate and use Suitable to operate in the liquid medium due to shear mode Better Q factor Low-cost device Better commercial availability	Miniature device with reduced sample volume Higher sensitivity due to higher operating frequency Operating frequency in the range of lower GHz to few tens of GHz Lower LOD in the range of picograms to femtograms Compatible with CMOS processing due to the use of piezoelectric thin films	Lower power consumption due to high surface energy density Low-cost device and easy to fabricate Higher sensitivity due to high operating frequency Operating frequency in the range of a few MHz to hundreds of MHz LOD in the range of nanomolar to hundreds of picomolar Wireless capability enabling POC devices Commercially available biosensors Ability to operate in liquid medium for all modes except R-SAW
Limitations	Lower sensitivity due to lower operating frequency Operating frequency in the range of 5 MHz to 30 MHz Comparatively lower LOD in the range of µg/mL Incompatible with CMOS processing Inability to batch produce due to bulk quartz substrate Difficult to scale down dimension	Difficulty in manufacturing fragile free-standing structures for FBAR with a back cavity Longer fabrication time and cost for SMR structures Not commercially made on a large scale Higher noise components while making measurements	Higher attenuation of acoustic waves in liquid medium for R-SAW Difficulty in exciting pure shear mode in SH-SAW and Love mode devices Impact of guiding layer to insertion loss for Love mode sensor Breakable structure for Lamb wave devices due to thin membranes

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
