# Peer review of "Acoustic Biosensors and Microfluidic Devices in the Decennium: Principles and Applications"

_micromachines, 2021, doi:10.3390/mi13010024_

Round 1

Reviewer 1 Report

Recommendation: Minor revision

In this review, the authors introduced the principals and applications of both acoustic biosensors and acoustofluidic devices. By presenting an extensive review of the use of acoustic waves in label-free biosensors, the authors discussed the mechanisms, applications, advantages, and challenges in both BAW and SAW biosensors. Additionally, the authors provided a brief overview to the acoustofluidic devices with their mechanisms and biological applications. In the end, the authors presented a perspective and summary of real time, label-free, portable LOC devices utilizing acoustic waves. Overall, this review is well written, and it could be a very useful guideline for researchers to investigate acoustic biosensors and acoustofluidic devices. However, I have a few concerns with some parts of the work which the authors should address before the paper can be accepted.

Major comments:

  • Organization: (a). In general, this review is well written, and the reviewer can easily follow it. However, the authors may considering reorganize the Acoustic Biosensors part with introduction of critical parameters, such as sensitivity, electromechanical coupling coefficient, quality factor, and temperature effect, in ahead, instead of presenting them only in BAW sensors part. (b). The reviewer appreciates the two tables provided in this review. The authors may consider adding a summary table that comparing the advantages and limitations for both BAW and SAW sensors. (c). The reviewer can understand that the authors try to introduce the detailed applications in each category of acoustic sensor, however, this provides more redundant information. The sensing of DNA, protein, virus, bacteria, and cell can all be introduced in an application part with a summary of advantages and limitations by comparing different kinds of acoustic sensors.
  • The idea of integration of both the acoustic actuation and sensing functions is sound. However, the authors provide very litter reference regarding this integration effort. With regard to the integration of acoustic actuation and sensing, some of the references should be discussed: 1. Biosensors and Bioelectronics, 2022 (196), 113730; 2. Small, 2020, 16(48), p.2005179.

Detailed comments:

  • The resolution of some figures, such as Fig. 5, Fig. 6, and Fig. 12, are quite low.
  • Fig. 2: The weight of lines/outlines in the top figure need to be decreased. For the bottom schematic, the meaning of the sine wave is not clear. Are the authors trying to illustrate the monitoring of frequency, amplitude, and phase as a sensor’s outputs?
  • Fig. 5: tq should be labeled in (a).
  • Line 275: The authors should double check the equation of (6).
  • Line 721: The authors should double check the equation of (11), especially the notation of f=f0.
  • Line 761: The authors have not defined the c axis. The authors should utilize uniform coordinates in the manuscript.
  • Line 1602: Compared with glass and silicon, the acoustic impedance of PDMS is relatively low. The reviewer doubts whether PDMS can be used as an effective resonant chamber.

Reviewer 2 Report

Dear authors,

Please consider the suggested comments to improve the quality of the current version of the manuscript:

  1. In the abstract, the synopsis of the review and the specific review point need to be addressed clearly. Also, the scope of the review needs to be established in detail. In the submitted manuscript the scope of the review is not clearly explained. Please confine the scope to a specific area and discuss/explain it.
  2. The review criteria is also very important in a review paper. In the abstract of the manuscript, the review criteria is not specifically mentioned. In a review paper, the scope of review and the review criteria are very critical.
  3. In the abstract please include the details of the contents and the specific points discussed in the review paper so that readers will have a clear understanding of the review paper. Providing the details of the contents of the review, in the abstract helps to provide necessary enthusiasm and interest to the readers and thus improve the number of full-text reads of the paper.
  4. The abstract also needs to incorporate an appropriate conclusion and quick thought inputs. In the current version of the manuscript, though the conclusions are mentioned (lines 19 & 22) but not detailed properly. Please update the abstract with detailed scientific conclusions.
  5. In the introduction of the manuscript, it is very important to explain the need for the review to be performed. In the current version of the manuscript, the need for this review is not clearly established. Please provide the justification of the review being performed in the manuscript as the topic of ‘Acoustic Biosensors’ and ‘Microfluidic Devices’ were widely studied over the last few years.
  6. Please also include the knowledge gaps in the studies performed by the peer researchers and the other review papers in this field. Kindly address those knowledge gaps in the manuscript. Very importantly, please specify the need for the current review presented in the manuscript.
  7. In the last paragraph of the introduction, include the details of the broader impacts on the study made, and address the future scope and topics that are important and could not be covered in this review.
  8. Please include a schematic with a tree diagram to show the classification of different acoustic biosensors made in the review process.
  9. Please include a table with details of the biosensors considered for the review, and the corresponding information of sensitivity, specificity, and limit of detection, etc., of those biosensors, to provide comprehensive information.
  10. Some of the references cited in the paper are not from recent years. Please include the studies performed and published more recently (preferably after 2018).
  11. Please include the appropriate thought inputs and prospects in the conclusions based on the review performed.
  12. Please revise the manuscript with English grammar. There are a few places that the manuscript needs to be improved with respect to English writing.

Round 2

Reviewer 2 Report

Dear authors,
Thank you for updating the manuscript with recommended changes.